# Spontaneous and Ionizing Radiation-Induced Aggregation of Human Serum Albumin: Dityrosine as a Fluorescent Probe

**DOI:** 10.3390/ijms23158090

**Published:** 2022-07-22

**Authors:** Karolina Radomska, Marian Wolszczak

**Affiliations:** Institute of Applied Radiation Chemistry, Faculty of Chemistry, Lodz University of Technology, 93-590 Lodz, Poland; karolina.radomska@dokt.p.lodz.pl

**Keywords:** pulse radiolysis, HSA aggregates, dityrosine, oxidative stress, free radicals

## Abstract

The use of spectroscopic techniques has shown that human serum albumin (HSA) undergoes reversible self-aggregation through protein–protein interactions. It ensures the subsequent overlapping of electron clouds along with the stiffening of the conformation of the interpenetrating network of amino acids of adjacent HSA molecules. The HSA oxidation process related to the transfer of one electron was investigated by pulse radiolysis and photochemical methods. It has been shown that the irradiation of HSA solutions under oxidative stress conditions results in the formation of stable protein aggregates. The HSA aggregates induced by ionizing radiation are characterized by specific fluorescence compared to the emission of non-irradiated solutions. We assume that HSA dimers are mainly responsible for the new emission. Dityrosine produced by the intermolecular recombination of protein tyrosine radicals as a result of radiolysis of an aqueous solution of the protein is the main cause of HSA aggregation by cross-linking. Analysis of the oxidation process of HSA confirmed that the reaction of mild oxidants (${\mathrm{Br}}_{2}^{•-}$, $N_{3}^{•}$, ${\mathrm{SO}}_{4}^{•-}$) with albumin leads to the formation of covalent bonds between tyrosine residues. In the case of ^•^OH radicals and partly, ${\mathrm{Cl}}_{2}^{•-}$, species other than DT are formed. The light emission of this species is similar to the emission of self-associated HSA.

## 1. Introduction

Protein aggregation in the aqueous solution can be initiated by various factors such as pH, temperature, or radicals, including those generated by ionizing radiation. The hydrogen bond networks of cross-β-sheet structure and/or π-π stacking interactions are the reason behind the origin of intrinsic blue autofluorescence in protein aggregates in protein solutions [1,2]. This type of aggregation is a reversible process; the α-helical structure of HSA is preserved without significant changes to the β-sheet structure. The concentration-dependent aggregates are structurally different from heat-denatured HSA aggregates.

The attack of radicals and reactive oxygen species on proteins results in the formation of albumin aggregates stabilized by intermolecular covalent bonds [3,4,5,6,7]. Dityrosine (DT), a highly fluorescent molecule, is widely regarded as a biomarker of oxidative stress, indicating protein damage due to radiation, aging, and exposure to free radicals including reactive oxygen and nitrogen species [8,9,10,11]. The oxidation of tyrosine leads to the formation of dityrosine, which was first described over 30 years ago [12]. DT can be found in several oxidatively modified proteins e.g., HSA, BSA, OVA, hemoglobin, insulin, lysozyme, myoglobin, trypsin or urease [13]. Dityrosine is formed as a result of the recombination of long-lived tyrosyl radicals generated under oxidative conditions. The formation of a stable carbon–carbon bond between tyrosine molecules leads to intra- or intermolecular cross-linking of proteins [14]. Increased DT levels play an important role in the pathogenesis of various diseases, such as eye cataracts, atherosclerosis, acute inflammations, Alzheimer’s or Parkinson’s disease, and skin cancer [15,16]. The overestimated proportion of albumin dimeric and oligomeric forms may be the first symptom of the development of neoplastic or neurodegenerative diseases.

Dityrosine and other products of tyrosine oxidation as biomarkers of protein oxidative modification have contributed to the development of methods for the analysis and quantification of these modifications. DT is commonly detected by high-performance liquid chromatography (HPLC) analysis and/or by fluorescence measurement. Although HPLC is a very sensitive technique, it does not provide structural data on the chemical link between Tyr molecules. The high sensitivity of fluorescence methods allows not only studying interactions in the protein solution but also facilitates the analysis of structural changes in proteins and the process of their aggregation. The method for measuring DT based on a proteolytic digestion followed by HPLC analysis with fluorescence detection was first described in detail by Davies et al. [13,17,18]. Dityrosine levels in mammalian tissues and urine samples were also measured by chromatographic separation followed by GC-MS and HPLC-MS/MS mass spectrometry [8].

It is reported in the literature that dityrosine formation is monitored by measuring fluorescence at 325 nm excitation and 400–410 nm emission. The appearance of a new “blue” fluorescence was observed in protein crystals and peptide fibers containing non-fluorescent amino acids [19,20]. It has been proposed that the new fluorescence is not related to the presence of aromatic amino acids in the peptide chain but rather results from electron delocalization through intramolecular or intermolecular hydrogen bond networks [21]. From a medical point of view, the blue fluorescence characteristic of protein aggregates may prove to be a more effective marker of amyloid than DT.

## 2. Results

### 2.1. Spontaneous Aggregation of Human Serum Albumin

The high sensitivity of fluorescence methods allows studying the protein structural changes and the process of their aggregation. The aromatic amino acids with intrinsic fluorescence properties: tyrosine (Tyr)-18, phenylalanine (Phe)-31 and only one tryptophan (Trp214) are responsible for the absorption and fluorescence of HSA in the near UV range. Although the fluorescence quantum yields of tryptophan and tyrosine are similar and several times higher than phenylalanine, the Trp214 residue plays a decisive role in the fluorescence of HSA. Both Tyr and Trp have the absorption maximum at 278 nm, but tryptophan, unlike tyrosine, shows significant absorbance at around 295 nm. For a wavelength of 295 nm, Trp214 can be excited without interference from the tyrosine. The contribution of Phe to the intrinsic fluorescence of HSA is negligible [22].

The emission spectra of neat HSA solutions in the concentration range of 30–300 µM has been registered to study the self-aggregation process of human serum albumin. The results of these experiments are shown in Figure 1 and clearly revealed that the light emission of protein aggregates depended almost linearly on the concentration of HSA. The excitation wavelength was 320 nm for exciting HSA self-aggregates. Our measurements do not support the thesis [1] that there is a concentration threshold (around 10 µM) above which self-aggregates of HSA are formed.

Helpful information on the process of albumin aggregation can be obtained by analyzing the fluorescence excitation spectra of the HSA solution. The fluorescence excitation spectrum of HSA (80 uM) in a buffer solution exhibits a maximum at 290 and 345 nm for the λ_em_ = 410 nm (Figure 2). The band centered at 290 nm can be attributed to the tryptophan residue Trp214. Analysis of the emission excitation spectra of neat HSA solution (80 µM) for λ_em_ = 450 nm suggest that in relation to the emission spectrum for λ_em_ = 410 nm, there is an additional, third band with a maximum near 390 nm. The clearly observed band at 290 nm indicates the inactivation of the excited state of the tryptophan molecule as a result of energy transfer from *Trp214 to HSA aggregates (FRET—Förster Resonance Energy Transfer). The excitation emission spectrum of the HSA solution shows two additional bands with a maximum at 350 and 390 nm, indicating the presence of two different types of aggregates. Time resolution measurements also show two main groups of aggregates. The excitation of HSA solutions with the laser light with a wavelength of 337 nm allows registration of the emission of excited protein aggregates. This light is not absorbed by monomeric HSA. Decay at 400 nm is characterized by an average lifetime close to 3 ns, for 450 nm, the analogous lifetime is about 4.2 ns. By changing the concentration of HSA in a wide range (30, 60, 80, 100, 150, 200, 300 µM), we registered slight changes in the lifetime of the emission of HSA aggregates as a function of protein concentration. For example, for the above-mentioned solutions, we recorded the average emission decay time for 450 nm, amounting to 4.18, 4.19, 4.3, 4.3, 4.49, 4.5, and 4.6 ns, respectively. Therefore, it can be postulated that in albumin solutions, even at a low concentration of HSA, there are species with different emission states. The new bands (at 350 and 390 nm) in the excitation spectrum are probably a consequence of the formation of the hydrogen bonds network and/or π-π interactions (van der Waals force) between HSA molecules.

The change in HSA concentration in the studied solutions causes changes in the intensity of the emission and excitation spectra, which seemingly are not a linear function of the protein concentration. As shown in Figure 1, the correction for the inner filter effect creates a linear relationship between the intensity of the emission band and the concentration of HSA over a wide range of HSA concentrations (30–300 μM).

The emission of light after the excitation of protein aggregates is caused by the coupling of non-aromatic chromophores, which ensures the subsequent overlapping of electron clouds along with the stiffening of the conformation of the interpenetrating network of amino acids of adjacent HSA molecules. In the scientific literature, there is no well-established term for this type of deactivation of excited states of aggregates. Typically, the aggregation of classic chromophore groups leads to the quenching of excited states (self-quenching). The reverse process of self-quenching, called aggregation-induced emission (AIE), occurs in some organic compounds [23]. In the literature, one can also find the term “blue autofluorescence in protein aggregates” [24] or intrinsic blue fluorescence from oligomeric interfaces of human serum albumin [1]. For the purposes of this work, we will call the light emitted by protein aggregates excited by light with a wavelength above 300 nm emission, not fluorescence.

We found a similar relationship for the excitation spectra of HSA for the concentration range of 30–300 μM of protein (see Appendix A in Appendix A). The fluorescence excitation spectrum of HSA neat solutions (30–300 μM) exhibits a maximum at ≈290 and 345 nm for the λ_em_ = 410 nm. The band centered at 290 nm can be attributed to the tryptophan residue Trp214, while the second band (λ_max_ = 345 nm) is characteristic of HSA aggregates. As the concentration of HSA increases, the value of the emission intensity increases in the maximum bands at 290 and 345 nm. The excitation spectra of albumin solutions confirmed that the intensity of emission of protein aggregates depends linearly on the concentration of HSA (it was confirmed by the dependence of emission intensity recorded and corrected at 345 nm as a function of HSA concentration).

The influence of oxygen on the HSA aggregation process is negligible. It is important to note that dilution of the albumin solution leads to a disaggregation of the HSA. The HSA solution (300 µM) was characterized by an intense spectrum with an emission band in the visible region with a maximum at 440 nm (λ_exc_ = 360 nm). The emission spectrum of the HSA solution with a concentration of 300 µM recorded after tenfold dilution is very consistent with the spectrum recorded for a solution of HSA with a concentration of 30 µM prepared by dissolving HSA in PBS solution. In this “dilution” experiment, the existence of non-covalent HSA dimers in equilibrium with monomeric HSA was demonstrated.

We have performed fluorescence measurements of aqueous solution of 1 µM peptide MD26 (Pro–Glu–Pro–Thr–Ile–Asp–Glu–Ser), and we noted no emission after excitation of the solution with the 295 nm light. After increasing peptide concentration to 100 µM, light emission was observed with a maximum at near 425 nm (λ_exc_ = 337 nm). The structure of MD26 (no aromatic residues) clearly indicates that hydrogen and ionic bonds were sufficient to form aggregates of emission-active polypeptides. In order to clarify whether the formation of spontaneous aggregates (self-aggregates associated with an increase in concentration of protein) is unique to HSA only, we performed comparative measurements for other proteins. The results will be presented soon in a subsequent publication. Here, for illustration, we will compile the emissions data collected for HSA with those recorded for BSA and papain. For each analyzed protein, there were self-aggregates in non-irradiated protein solutions. The excitation of protein solutions with the 320 nm light results in the appearance of emissions at 410 nm (one band centered at 415 nm was observed for the papain solution). The excitation spectrum of the 430 nm emission band shows a sharp peak at 292 nm (Trp214) and around 350 nm (aggregates). It means that the tryptophan residue and the newly formed chromophore groups are close to each other. After filtering the protein solution, no changes in the emission and excitation spectra were observed, which proves the high stability of the self-aggregates (see Figure 3 data presented only for HSA). The fluorescence lifetimes of HSA aggregates (C_HSA_ = 1.6 mM) varies between 4.3 and 4.75 ns, depending on detection wavelength. The emission lifetime values were calculated with the mono-exponential decay. Increasing the detection wavelength to 480 nm leads to an increase in the lifetime of the fluorescence of HSA aggregates. Above this value (480 nm), the lifetime of the HSA aggregates emission slightly decreases. The determined emission lifetime values confirmed that two populations of protein aggregates are presented in the concentrated HSA solution, which can be distinguished on the basis of time-resolved fluorescence measurements. The use of a more accurate measurement technique based on SPC single photon counting should help distinguish protein aggregates obtained by the oxidation of albumin molecules with azide or hydroxyl radicals. For accurate analysis of the emission of HSA aggregates, we always perform a series of measurements for different excitation and emission wavelengths and do not use only the DT fluorescence detection procedure (λ_exc_ = 320 nm; λ_det_ = 410 nm).

Using spectroscopic techniques, we have confirmed that HSA, BSA and papain undergoes reversible aggregation as a result of protein−protein interactions. Moreover, we have shown that two types of self-aggregates are formed. In our opinion, dimers are mainly responsible for blue emissions. We assume the possibility of formation of different dimer types (depending on the structural changes of the protein). Consequently, each solution will have a complex equilibrium between many variants of HSA dimers, which is not described by a unique equilibrium constant. Electrophoretic analysis confirmed that the presence of dimers, trimers and tetramers in the solution increases with increasing HSA concentration (described further). We called these higher-than-dimeric forms of HSA complexation spontaneous aggregates. It is also known from HPLC measurements that dimers and aggregates constitute about 16 and 7% of all HSA molecules in the non-irradiated albumin solution, respectively (concentration of HSA 15 µM) [25,26]. Unfortunately, fluorescence measurements cannot be used directly to estimate the percentage of dimers and other oligomers in the HSA solutions. It is due to the unknown value of the fluorescence quantum yield of HSA aggregates. It is not easy to determine the fluorescence quantum yield of protein aggregates due to their different structures (dimers, trimers, etc.) and scattering light in concentrated HSA solutions.

Our experiments confirmed also that pH has an influence on the aggregation process of HSA. The results show that the decrease in pH from 7.0 to 5.0 gradually unfolded the HSA structure to slightly increase the emission intensity of HSA aggregates in the aqueous solution.

An interesting observation from our experiments is quenching of the excited state of HSA aggregates by *N,N’*-dimethyl-4,4′-bypyridinium (methylviologen; MV^2+^). Fluorescence quenching of the excited state of aggregates by the MV^2+^ has been studied by steady-state and time-resolved techniques. The quenching data were analyzed by using a conventional Stern–Volmer equation. On the basis of steady-state emission measurements and emission lifetime measurements of HSA aggregates with the addition of MV^2+^ in various concentrations (0–90 mM), a Stern–Volmer plot was made. The SV dependence is shown in Figure 4. The I_0_ and I value for a given quencher concentration were read from the emission spectra recorded in the absence and presence of the methylviologen. The Stern–Volmer plot presented in Figure 4 indicates a good linear relationship in the quencher concentration range up to 20 mM. Above this concentration value, the graphs deviate from linearity. It proved both static and dynamic quenching. The sublinear relationship of the Stern–Volmer curve was related to the formation of the MV^2+^–HSA aggregate complex. 

### 2.2. Influence of Temperature on the Process of Self-Aggregation of HSA

The emission of HSA solution (600 µM) has been measured for the temperature range 20–70 °C. Figure 5 presents the fluorescence emission spectra of buffer HSA solution for λ_em_ = 345 and 500 nm. The excitation spectrum of HSA solution was characterized by three bands centered at 290, 360 and 400 nm. The increase in the temperature of the solution leads to decreasing the emission quantum efficiency of the *Trp214 (λ_em_ = 345 nm) compared to the solution at 20 °C, which is typical for classic luminophore. The decrease in the intensity of HSA emission is due to nonradiative transitions as the temperature continues to rise. The gradual increase in the temperature of the HSA solution leads to a decrease in the emission quantum yield of protein aggregates (λ_em_ = 500 nm). The decrease in emission intensity with increasing temperature was almost monotonic for the excitation band I for the temperature range 20–70 °C. On the other hand, the emission intensity of excitation band II slightly increased above 50 °C, which confirmed that two populations of aggregates are present in the HSA solution. Higher temperature promoted the formation of one of them: a β-sheet structure stabilized by hydrogen bonds. Following heating, dimer formation is observed prior to protein melting, while no higher-order aggregates are observed in the 20–60 °C temperature range. It was postulated that the lack of a further decrease in the emission intensity with increasing temperature for the excitation band I suggests the presence of “dynamic” aggregates. As dimer and aggregates formation results in a decrease in the accessible protein surface, the number of water molecules in the hydration shell of the protein per monomer becomes lower at higher temperatures [27].

The gradual heating and subsequent cooling of the albumin solution did not significantly affect the emission spectra of HSA, which confirmed that the conformational changes of the protein are reversible at temperatures below 70 °C (HSA particle diameter was about 10 nm). Self-aggregates of albumins are chemically different from those thermally generated. Aggregates obtained by heating HSA solutions to temperatures above 70 °C undergo a conformational change, the so-called unfolding. Albumin aggregates are surrounded by about three layers of water. Heating the protein solution also changes the hydration of the albumin, particularly affecting the first layer of water molecules surrounding the HSA. If warm solutions are kept on a scale of minutes or hours, their natural structure is not recovered by cooling as a result of penetrating and overlapping the neighboring protein molecules. The protein–protein electrostatic interactions of HSA play a predominant role in thermal-induced aggregate formation. The mechanism of heat-induced protein aggregation proceeds via two different pathways: the formation of small “soluble” aggregates by the conversion of α-helix to β-aggregated structures as a result of electrostatic interactions or the formation of aggregates of larger size via hydrophobic interactions.

### 2.3. Albumin Aggregation in Deuterium Oxide Solutions

Deuterium oxide (D_2_O), widely used in protein characterization studies, has been shown to promote protein aggregation when used as a substitute for water in most buffered protein solutions [28,29]. Lee and Berns reported that D_2_O enhances hydrophobic forces thought to contribute to the formation of the protein aggregates [30]. Recent studies demonstrated that exchanging water with D_2_O can improve the stability of proteins in solution by maintaining the stability of the monomeric form [31]. This effect was predominantly due to an enhancement of hydrophobic interactions with possible contribution from an increase in the strength of intra- or intermolecular bonds due to hydrogen–deuterium exchange.

The emission spectrum of HSA aggregates in a D_2_O solution is slightly red shifted compared to buffer solution (Δλ = 2 nm). The emission measurements confirmed that D_2_O did not inhibit the aggregation process of HSA. Our experiments demonstrated that exchanging water with D_2_O leads to an increase in the emission intensity of HSA aggregates (Figure 6). The normalized emission and excitation spectra of HSA aggregates showed no differences, which may indicate that in the case of D_2_O, only the increase in fluorescence quantum yield of HSA is observed.

### 2.4. Effect of Solvent on HSA Aggregation

There is a vast literature on the influence of solvents on the protein aggregation process and thereby on the stability of proteins. The protein molecules are surrounded by a hydration layer, which protects them against spontaneous aggregation in aqueous solutions. The structure and dynamics of the inner hydration layers of proteins are affected by hydrophobic interactions, hydrogen bonds, electrostatic interactions and van der Waals forces. Organic solvents including alcohols also promote the aggregation of albumin. In this work, we have investigated the aggregation of human serum albumin in various solvents. Figure 7 shows the emission and excitation spectra of HSA recorded in various solvents: *t*-BuOH: PBS 1:1, D_2_O, EG:H_2_O 1:1, EG:H_2_O 1:1 in 77 K. The results obtained from detailed investigation have shown that the aggregation of HSA is favored in the above-mentioned solvents. In all solutions of HSA, an increase in the emission intensity of protein aggregates was observed in relation to the albumin buffer solution. It is important to note that freezing (77 K) the concentrated albumin solution (300 uM) significantly shifts the emission spectrum of HSA toward shorter wavelengths. This spectral change is due to the stiffening of the HSA structure in 77 K. Two populations of HSA aggregates are observed in the 77 K matrix (excitation band with a maximum at 360 and 400 nm). The formation of two types of aggregates was also observed in other solvents. The addition of *t*-BuOH enhanced the formation of aggregates characterized by a band with a maximum near 400 nm. After a few hours from the preparation of the HSA solution containing *t*-BuOH, we observed the precipitation of protein deposits from the solution. The detailed analysis shows small changes in the HSA spectra regardless of the type of solvent beside matrix in 77 K. 

The physical processes of aggregation, especially in salt solutions, are poorly understood. Protein aggregation has been studied as a function of the types and concentrations of added salt in the HSA solutions: NaClO_4_ (2 M, 7 M), NaSCN (2 M) and NaI (2 M). Our study of the protein aggregation process confirms that it is modified by the presence of salt in the solution. Regardless of the type of added salt to the HSA solution, we observed a significant decrease in the intensity of albumin aggregate emission for various emission and excitation wavelengths (Figure 8 shows the emission and excitation spectra of HSA solution after excitation with 360 nm light and detection at 450 nm, respectively). The significant changes in the emission intensity of HSA aggregates have been observed in the case of protein solution containing NaI. These spectral changes suggest quenching of the excited state of HSA aggregates in the presence of a heavy atom (classic quencher of *Trp) [32]. This is due to the formation of singlet exciplex, which rapidly dissociates as a result of the remarkable enhancement of intersystem crossing with the heavy-atom substitution.

It has been shown that the emission of HSA aggregates characterized by an emission band with maximum near 400 nm is efficiently quenched in this case (Figure 8). It means that iodide anion is in close proximity to the newly formed chromophore groups. but probably does not modify the structure of a single protein molecule. The addition of NaSCN also leads to a decrease in the intensity of emission of HSA aggregates, which means that the presence of NaSCN significantly influences the HSA aggregation process. If the concept of inhibiting the process of HSA aggregation/destabilization of aggregates by NaSCN was correct, it could be assumed that electrostatic interactions are also involved in the aggregation process [33]. The reason for the decrease in the intensity of emission of HSA in the case of the remaining salts used in the experiment was probably a decrease in the fluorescence quantum yield of the HSA aggregates in their presence. After normalizing the HSA excitation spectra, it can be concluded that both types of HSA aggregates are quenched in the presence of salts.

The aggregation behavior of HSA in the presence of anionic surfactant sodium dodecyl sulfate (SDS, 20 and 60 mM) and the cationic surfactant dodecyltrimethylammonium chloride (DTAC, 60 mM) has been studied. The process of aggregation was found to be promoted by the electrostatic binding between the peptide and surfactant monomer at the concentration below CMC. However, increasing the surfactant concentration gave rise to a repulsive force among the head groups of surfactants bound on the peptide and the generation of a hydrophobic cluster of surfactant molecules leading to sheet structure destruction [34]. Our experiments indicate that surfactants did not inhibit the HSA aggregation process. At low concentrations of surfactants, where its contribution to the overall charge is modest, protein aggregation has been observed. In HSA solution containing a high concentration of SDS (60 mM), the positive charges of the surfactant outnumbered the negative charges of HSA, and therefore, the repulsive electrostatic interaction became dominant, as in the case of DTAC solution. This resulted in a decrease in the intensity emission of HSA aggregates (about 12.5% from the initial emission). The presence of surfactants also solubilizes unfolded protein monomers, leaving them unavailable for protein–protein association and thus inhibiting aggregation.

### 2.5. Influence of HSA Aggregates on the Structure of Water

The three-dimensional structure of water is extensively studied not only in neat water but also in the presence of biologically important molecules e.g., proteins [35]. In liquid water, a network of H-bonds is established, resulting in a highly complex molecular structure. This structure leads to many overlapping bands in the IR spectrum. The four major water NIR absorption bands are centered near 970, 1200, 1450 and 1900 nm [35]. Strong near-infrared absorption bands of water near 1400–1440 nm have often been applied to a quantitative analysis of water content in foods and pharmaceuticals. Figure 9 shows the NIR absorption spectrum of HSA solutions (15 μM–1.8 mM) in the region of 1350–1650 nm. The importance of the NIR spectrum of water results from the fact that the intensity of the band due to water alters with changes in the strength of hydrogen bonds. Contrary to concentrated albumin solutions (over 600 μM), no changes in the water structure were observed in the presence of HSA at the physiological concentration (600 μM).

Near-infrared (NIR) spectroscopy was used to examine the properties of water in several concentrated salts solutions containing HSA (300 µM). It is shown that NaClO_4_ substantially changes the water structure, whereas NaI and NaSCN influence the water structure insignificantly. In the case of NaClO_4_, the absorption spectrum recorded in the spectral range 1300–2100 nm is dramatically different with respect to the analogous spectrum recorded in neat HSA solution. The addition of NaClO_4_ results in a shift of the bands with a maximum at 1450 and 1930 nm toward shorter wavelength and produces a shoulder around 1462 nm (Appendix A, Appendix A). The presence of NaI and NaSCN changed the intensity of absorption bands centered at 1450 and 1930 much with respect to the neat water absorption spectrum. The spectral changes of NIR absorption bands in the salts solutions containing HSA are due to the modification of the water structure.

### 2.6. Radiolysis of Tyrosine Aqueous Solution

To study the reaction of ^•^OH radical with HSA we applied pulse radiolysis. Figure 10 shows transient absorption spectra recorded at various times after the electron pulse irradiation of N_2_O-saturated aqueous solution containing 2 mM Tyr. The broad absorption band centered at 330 nm was observed immediately after irradiation. This spectrum can be attributed to the absorption of the OH-adduct (hydroxycyclohexadienyl-radical) on the ortho-position to the OH group. New absorption in the visible region with a maximum near 408 nm is due to the phenoxyl radical of tyrosine: TyrO^•^. The main process of reaction of a hydroxyl radical with tyrosine is the formation of an ortho-directed OH adduct (band with a maximum at 330 nm) [36]. Our measurements are in good agreement with the literature. 

The rate constant of this reaction is equal (7.0 ± 0.5) × 10^9^ dm^3^ mol^–1^ s^–1^ [37]. The ortho-directed OH adduct decays by water elimination according to a first-order reaction under the formation of phenoxyl radical (2k = 3.0 × 10^8^ dm^3^ mol^–1^ s^–1^). The phenoxyl radical, TyrO^•^, is additionally formed (5%) as a result of a hydroxyl radical direct reaction with the OH group of Tyr (H-abstraction). The rate constant of this reaction is equal to (6.0 ± 1.0) × 10^8^ dm^3^ mol^–1^ s^–1^ (absorption band with maximum at 408 nm) [38,39]. The meta isomer of the OH adducts has two absorption maxima at 305 nm and 540 nm [37]. Figure 10 also shows the transient absorption spectrum of oxidized tyrosine molecule by azide radical. The transient absorption spectrum recorded during the pulse radiolysis of 2 mM Tyr solution containing 0.1 M NaN_3_ shows maxima at 305 and 408 nm. A reaction of azide radicals with Tyr leads to the formation of TyrO^•^.

Steady-state radiolysis was applied to study the influence of ionizing radiation on the absorption spectra of Tyr solution. The absorption spectra of the N_2_O-saturated buffer solution containing 2 mM Tyr (10 mM phosphate buffer pH 7.2) in the suprasil quartz cell were recorded after irradiation with doses between 0 and 1300 Gy and are presented in Figure 11. Before irradiation, the absorption spectrum of Tyr in the presented region shows one band with a maximum at 275 nm. The molar absorption coefficient of this band is ε_275nm_ = 1410 M^–1^ cm^–1^. On irradiation of this solution (exposed to multiple electron pulses), there is a change in the intensity of the band, and a new absorption band develops in the spectral range 300–350 nm. New absorbance can be attributed to the dityrosine formed as a result of the oxidation of Tyr molecules with ^•^OH radicals. The spectrum of DT was obtained by subtracting the Tyr spectrum before and after irradiation of the solution (Tyr_1300 Gy_ − Tyr_0 Gy_). The DT concentration in irradiated tyrosine solutions was monitored by UV-ViS spectroscopy. The amount of DT generated in the reaction of Tyr with the ^•^OH radical was estimated on the basis of absorption spectra. The molar absorption coefficient of DT ε_315 nm_ = 6500 M^–1^ cm^–1^ was determined by Ionescu et al. [40]. Analogous calculations were made for the N_2_O-saturated tyrosine solution with the addition of 0.1 M NaN_3_. As a result of the hydroxyl radicals reaction with tyrosine, 60 μM DT (A_315 nm_ = 0.32) is formed, while in the case of azide radicals, 12 μM of dityrosine (A_315 nm_ = 0.07) was generated.

The impact of pH on the process of tyrosine formation is pronounced. Steady-state radiolysis measurements of N_2_O-saturated Tyr solutions were performed in the range of pH 2–11. The emission spectra of the solutions containing 2 mM tyrosine at various pH after irradiation were recorded and are presented in Figure 12. The excitation of aqueous Tyr solution at pH 7.3 and 11 with the 320 nm light leads to developing an emission band centered at 405 nm, as shown in Figure 12. The highest yield of DT emission was observed for the solution at pH 11. In an acidic environment, the quantum efficiency of DT emission is the lowest.

The emission excitation spectra (Figure 12) of tyrosine recorded for the same solutions differ significantly. Tyrosine molecules excited in neutral aqueous solutions show the fluorescence of both the acidic and alkaline form of Tyr. It is due to the competitive process of Tyr excited-state deprotonation to emission and nonradiative transitions. DT can exist in both acidic and alkaline form, depending on the pH of the solution (pKa = 7.25). At physiological pH, more than 50% of DT exist in alkaline form [41]. Dependence of the DT emission on absorbed dose by aqueous solutions at various pH was observed. The linear dependence of intensity of emission as a function of the absorbed dose suggests that phenoxyl radicals generated in pulse radiolysis measurements recombine intermolecularly and form DT. The literature does not describe the dependence of the DT emission intensity obtained by radiation as a function of the dose absorbed by tyrosine solutions of different pH. Absorption spectra recorded after electron beam irradiation of HSA solutions and γ radiation show that DT is more efficiently generated in a solution exposed to γ radiation. This is due to the recombination of part of azide radicals in pulse radiolysis measurements. The pulse radiolysis of N_2_O-saturated aqueous solutions containing high concentrations of NaN_3_ (1M) has been found to produce $N_{6}^{•-}$ [42]:(1)$$N_{3}^{-}+N_{3}^{•}\to N_{6}^{•-},$$

In contrast to pulse radiolysis, the gamma irradiation of a solution containing NaN_3_ does not lead to a reaction between azide radicals and azide ions.

### 2.7. Radiolysis of Solutions Containing Tyrosine and Additionally One or More Selected Amino Acids 

The process of DT formation in irradiated solutions containing tyrosine and additionally one or more selected amino acids was analyzed. The reactivity of oxidative radicals with peptides depends on the constant rate of reaction of these radicals with the individual amino acids that make up the peptide. The aim of our stationary measurements was an attempt to answer the question of whether in a solution containing various amino acids it would be possible to observe a different fluorescence than DT after excitation with 320 nm light. Pulse radiolysis of saturated N_2_O solutions containing Cys and Tyr in the proportions 1:1 and 5:1 was performed to verify that the irradiated amino acid solutions would give an emission spectrum with a maximum other than about 403 nm (DT band). Our experiments allowed establishing that the possible generation of Cys–Tyr bonds cannot be responsible for the new blue emission of irradiated HSA solutions (different from DT). Steady-state radiolysis experiments of buffer solutions containing aromatic amino acids—Tyr, Trp, Cys, Met (solution A)—and solution containing Tyr, Trp, Met, Ser, Leu, Lys, and Arg (solution B) were carried out. Regardless of the type of amino acids (aromatic, sulfur-containing or other amino acids within HSA), irradiation of these solutions in the presence of oxidizing radicals leads to the generation of DT. The reaction of amino acids with azide radicals leads to the formation of more DT compared to analogous reactions with hydroxyl radicals. Another important observation from steady-state radiolysis measurements was the presence of DT in the irradiated peptide solution of VEALYL (Ala-Glu-Ala-Leu-Tyr-Leu). The results of the experiments described in this paragraph clearly revealed that the generation of “mixed” dimers, for example those generated as a result of recombination between different amino acids radicals (Cys and Tyr or Tyr and Trp), does not result in emission in the visible range.

### 2.8. Radiolysis of Solutions Containing HSA

We also measured changes in absorption and emission spectra related to steady-state irradiation for HSA solutions. The absorption spectra of a saturated N_2_O aqueous solution containing 30 μM HSA were recorded after irradiation with doses from 0 to 7200 Gy and shown in Figure 13. The absorption spectrum of HSA before irradiation shows two bands, one with a maximum at 280 nm. The molar absorption coefficient of this band at 280 nm is equal to 35,500 M^−1^ cm^−1^ [43] and the other monotonically rises to far UV. The molar absorption coefficient at 210 nm was estimated to be approximately 10^6^ M^−1^ cm^−1^ [43]. The absorption band peaking at 280 nm is mainly a consequence of the π→π * electronic transitions of two aromatic amino acids: Trp and Tyr residues. After irradiating this solution with multiple electron pulses, the band intensity (maximum at 280 nm) grows, and a new absorbance appears in the spectral range of 300–400 nm. The irradiation of HSA solution caused a marked increase in absorbance at 278 nm (A_800 Gy_ = 1.18→A_7200 Gy_ = 1.89). The new absorbance (between 300 and 400 nm) is attributed to the formation of HSA aggregates during oxidative stress. As seen from the insert of Figure 13, the absorbance value increases in proportion to the absorbed dose by the albumin solution. The dependence of absorbance as a function of dose is linear, regardless of the analyzed wavelength (Figure 13 shows dependence of A_278 nm_ and A_320 nm_ as a function of absorbed dose).

The newly formed absorbance (dash line in Figure 13, dose absorbed 7200 Gy) is the result of light absorption by the generated HSA aggregates and is partly the result of light scattering by the solution. 

It is reported in the literature that dityrosine formation is monitored by measuring the fluorescence at 325 nm excitation and 400–410 nm emission e.g., 400 nm [44], 409 nm [45], and 410 nm [46]. The excitation of irradiated aqueous HSA solution with the 320 nm light leads to the emission of DT centered at 420 nm, as shown in Figure 14. The emission intensity of HSA aggregates formed as a result of the reaction of albumin with ^•^OH radicals depends linearly on the absorbed dose (regardless of the excitation wavelength). HSA aggregates generated in the irradiated solution differ from self-aggregates (Figure 14 shows the emission spectrum recorded for a concentrated HSA solution, 300 µM).

In the case of electron beam irradiation of HSA solution (30 µM) containing NaN_3_ (0.1 M), the absorbance value and emission intensity increase monotonically to absorbed doses but non-linearly (Figure 15, insert). The aggregates formed in the reaction of HSA with azide radicals were characterized by classic DT fluorescence (the emission spectrum recorded after excitation of buffer HSA solution with 320 nm light shows one band with maximum near 405 nm). 

The conducted experiments confirmed that the mechanism of the azide and hydroxyl radical reactions with albumin differs significantly (Figure 16). The hydroxyl radical often reacts simultaneously via several competing pathways, e.g., H atom abstraction, addition to a double bond or to an aromatic ring, or via electron transfer; therefore, a number of intermediates may initially be present. Contrary to hydroxyl radical, $N_{3}^{•}\;$radicals react mainly by one-electron transfer and are much more selective than ^•^OH. The reaction of ^•^OH with HSA is leading to a fluorescent product with a maximum at 420 nm. Figure 16 shows the broad band due to the superposition of emission of DT and HSA aggregates (formed from different dimers: fluorescent and non-fluorescent). In the case of reaction of $N_{3}^{•}\;$radical with HSA, we observed the band characteristic of dityrosine. The diffusion of the azide radicals to the protein interior is postulated. The scavenging of $N_{3}^{•}\;$radicals by tyrosine residue within protein is a complex process. The HSA contains 17 tyrosine residues in a single polypeptide chain. The reactivity of azide radicals toward these Tyr residues differs significantly. Moreover, not every “meeting” of azide radicals with the tyrosine residue in the HSA structure leads to the formation of DT, and not every modification of tyrosine results in the formation of fluorescent products. According to the literature, the reaction of two TyrO^•^ radicals generated in different HSA molecules does not always lead to the formation of fluorescent tyrosine dimer [47]. The overlapping of the emission bands of both DT and self-aggregates makes it very difficult to identify the ortho-tyrosine dimer in optical measurements.

The analysis of the HSA oxidation process with the use of a number of radiation-generated oxidants was performed. Combined pulse radiolysis and enzyme-activity studies allowed determining which amino acid residues within biomolecules are modified as a result of radical attack. Identification of the nature of damage caused by hydroxyl radicals responsible for the inactivation of proteins or enzymes was difficult due to the lack of selectivity of ^•^OH radicals in reactions with amino acids. For more detailed analysis of the modification of amino acid residue structures, specific inorganic anion radicals were used. Pulse radiolysis and enzyme-activity studies have shown that the radiation-induced inactivation of lysozyme in neutral aqueous solution is due principally to the reaction of hydroxyl radicals with tryptophan residues [48]. In the case of trypsin, histidine damage leads to inactivation of the enzyme [49]. The methodology described above (pulse radiolysis with simultaneous analysis of enzymatic activity) allowed determining which amino acid residues are modified in the presence of various oxidizing radicals [50,51]. 

The position of individual amino acids in the polypeptide chain within the protein is important in analyzing the oxidation and reduction process of albumin. Analysis of the crystal structure of HSA revealed that 40% of the aromatic amino acids are close to the surface in the native protein structure [52]. Human serum albumin contains 31 phenylalanine residues, 1 tryptophan and 18 tyrosine residues in the peptide chain. Human serum albumin contains 31 phenylalanine residues, 1 tryptophan and 18 tyrosine residues in the peptide chain. In publication [53], the quantum yield of singlet oxygen generated by excited states of aromatic amino acids included in the structure of HSA (Trp, Tyr, Phe) were determined by time-resolved phosphorescence measurements. Human serum albumin has a total of 21 surface exposed residues, 11 partially occluded (5 tyrosine and 6 phenylalanine residues), and 19 residues that are buried deeply within the HSA structure. If only surface residues could sensitize oxygen, the quantum yield of singlet oxygen would be smaller than that obtained with an equivalent number of free aromatic amino acids in solution. The expected quantum yield of singlet oxygen would be about 40% of the weighted average quantum yield. The determined value of quantum yield of singlet oxygen experimentally is in very good agreement with the assumption that readily available light-excited HSA aromatic residues are involved in the reactions with oxygen molecules. The close correlation between the efficiency of singlet oxygen and the availability of aromatic residues in the HSA structure for oxygen molecules is for us the most interesting aspect of the discussed publication. By using different oxidizing radicals in our pulsed radiolysis measurements, we can expect that certain aromatic groups of HSA will be more readily available for attack by the radical. In our publication on the reaction of reducing radicals ($e_{\mathrm{aq}}^{-}$, H^•^, ${\mathrm{CO}}_{2}^{•-}$) with HSA, we have shown that the Sudlow 1 site, and in particular Trp214 located here, is well protected against these radicals [54]. Additionally, by generating radicals with different redox potential, we can to some extent decide which of the aromatic residues will be involved in the radical scavenging. 

It was postulated that the amino acid residues that are modified by ^•^OH radicals were located on the protein surface regardless of the pH of the solution [55]. Hydroxyl radicals react not only with aromatic amino acids but practically with all amino acid residues on the protein surface due to their high reactivity. It is known that ^•^OH radicals could generate radicals centered on the α-carbon atom and 27% of the total amount of radiation generated hydroxyl radicals in the BSA solution reacts with the protein, resulting in the formation of new carbonyl groups [55].

Pulse radiolysis results provide helpful information on inter- and intramolecular electron transfer in protein solutions. Intramolecular long range electron transfer (LRET) in the hen egg-white lysozyme accompanying radical transformation Trp^•^→TyrO^•^ was observed [56]. In this study, the authors observed LRET between the following amino acid residues: Trp62/Tyr53, Trp63/Tyr53 and Trp123/Tyr23. On the other hand, HPLC and MALDI-TOF MS measurements that were made for a solution of lysozyme containing NaN_3_ (0.1M) and γ irradiated indicate that Trp108 and/or 111 remain oxidized and that Tyr20 and 53 give DT [57]. There is no evidence for the reaction of tyrosine at position 23 with $N_{3}^{•}$ radicals, so LRET does not always take place in the protein structure. Other amino acids were probably involved in electron transfer (which are not always recorded in pulse radiolysis due to measurement difficulties, e.g., low molar absorbance coefficients).

Figure 17 shows the excitation spectra recorded after pulse radiolysis of a neat HSA solution (70 µM) and solutions containing additionally NaCl, LiBr or NaN_3_. All solutions were saturated with N_2_O. The salt concentration of the HSA solutions was 0.1 M. In the next stage, deaerated HSA solution (70 µM) with the addition of 5 mM (NH_4_)_2_S_2_O_8_ and 0.1 M *t*-BuOH was irradiated. The water radiolysis results in the formation of three well-characterized reactive radical species used to initiate radical reactions: $e_{\mathrm{aq}}^{-}$, ^•^OH and H^+^. A system containing only hydroxyl radicals can be obtained by saturating the aqueous solution with N_2_O, while the hydrated electrons are converted into the ^•^OH radicals. The azide radicals were formed in irradiated N_2_O-saturated aqueous solution containing NaN_3_. The sulfate radical anions were formed in the reaction of $e_{\mathrm{aq}}^{-}$ with (NH_4_)_2_S_2_O_8_. The dihalide radicals, ${\mathrm{Br}}_{2}^{•-}$, ${\mathrm{Cl}}_{2}^{•-}$ can be generated by pulse radiolysis carried out in LiBr or NaCl N_2_O-saturated aqueous solution (hydroxyl radicals reacts with halide ions to form oxidizing species). The pulse radiolysis of aqueous solution allowed producing reactive oxidizing radicals, such as ^•^OH, ${\mathrm{Br}}_{2}^{•-}$, ${\mathrm{Cl}}_{2}^{•-}$, $N_{3}^{•}$ and ${\mathrm{SO}}_{4}^{•-}$ formed via the following reaction:(2)$${\mathrm{Br}}^{-}|{\mathrm{Cl}}^{-}+{\mathrm{HO}}^{•}↔{\left(\mathrm{BrOH}|\mathrm{ClOH}\right)}^{-•},$$
(3)$${\left(\mathrm{BrOH}|\mathrm{ClOH}\right)}^{-•}\;\to {\;\mathrm{Br}}^{•}+{\;\mathrm{OH}}^{-},$$
(4)$${\mathrm{Br}}^{•}|{\mathrm{Cl}}^{•}+\mathrm{Br}|{\mathrm{Cl}}^{-}↔\mathrm{Br}|{\mathrm{Cl}}_{2}^{•-},$$
(5)$$e_{\mathrm{aq}}^{-}+{\;S}_{2}O_{8}{}^{2-}\;\to {\;\mathrm{SO}}_{4}^{•-}+{\;\mathrm{SO}}_{4}^{2-},$$

The HSA solutions containing NaCl, LiBr or NaN_3_ irradiated with a dose of 420 Gy were analyzed spectroscopically by measuring the absorption and emission spectra. The emission spectra recorded for the excitation wavelength 320 nm before and after irradiation of the solutions are presented in Figure 17. The emission spectra of irradiated HSA solutions differ markedly depending on the type of oxidant used. As a result of the reaction of radicals ${\mathrm{Br}}_{2}^{•-}$, $N_{3}^{•}$, ${\mathrm{SO}}_{4}^{•-}$ with HSA, mainly tyrosine residues are modified and DT is formed, which is manifested by an intense emission band with a maximum of about 400 nm (after excitation of irradiated solution with the 320 nm light). A similar conclusion regarding the formation of DT was drawn in the case of irradiation of lysozyme oxidized with ${\mathrm{Br}}_{2}^{•-}$ radicals [58,59]. Consequently, of the reaction of anion radicals ${\mathrm{Br}}_{2}^{•-}$ with lysozyme, dimers of enzyme are formed, which are characterized by emission centered at about 400 nm (the dimer is formed as a result of the intermolecular recombination reaction of phenoxyl radicals). Albumin oxidation with radicals ${\mathrm{SO}}_{4}^{•-}\;$ also leads mainly to the formation of DT bridges. In this case, the emission spectrum is broader in comparison to the DT band. It is due to the superposition of fluorescence of dityrosine and the emission of aggregates formed by the network of hydrogen and/or ionic bonds, analogous to self-aggregates (Figure 17).

The generation of ${\mathrm{Br}}_{2}^{•-}$ leads to a greater number of dityrosine bridges compared to DT obtained by other investigated oxidizing radicals. The emission intensity of HSA aggregates obtained in solutions in which the hydroxyl radical or chloride radical anion was generated is much lower compared to the emission of aggregates formed in the reaction of HSA with other oxidants. The more detailed analysis shows a red shift in the maximum of the band after the irradiation of N_2_O-saturated HSA (^•^OH) solution (403 nm→422 nm, ΔE = 0.1385 eV). The reaction of the${\;\mathrm{Cl}}_{2}^{•-}$ radical with HSA leads also mainly to species other than DT, as evidenced by a broad, less intense slightly red-shifted band (λ_max_ = 408 nm, Figure 17). This means that in the case of reaction of ^•^OH or ${\;\mathrm{Cl}}_{2}^{•-}$ radicals with HSA, aggregates were formed that were not stabilized by DT bridges. The high reactivity of the ^•^OH or radical ${\;\mathrm{Cl}}_{2}^{•-}$ with amino acids contained within HSA is responsible for the formation of various covalent bonds (including C-C and S-S) between adjacent protein molecules [60]. Hashimoto et al. explain the formation of DT in lysozyme solutions as a result of the reaction of this protein with ${\mathrm{Br}}_{2}^{•-}$ radical anion. It was also reported that in the reaction of lysozyme with hydroxyl radicals, the tryptophyl residues are the main sites to be attacked by the radicals [58]. In a pulsed radiolysis experiment of lysozyme solution, the authors observed an absorption band around 400 nm typical for the phenoxy tyrosine band. It has been postulated that the intramolecular transfer of the radical site from the tryptophyl radical to the tyrosyl radical occurs in a lysozyme solution. This argument was confirmed spectrophotometrically; the lysozyme dimers produced by the ${\mathrm{Br}}_{2}^{•-}$ radicals show a similar fluorescence spectrum to dityrosine [58]. On the basis of the reactivity of amino-acid residues located on the surface of lysozyme, the authors estimated that about 50% of the hydroxyl radicals attack aromatic amino acid residues, and 60% of these ^•^OH radicals (i.e., 30% of all OH radicals) react with tryptophan residues. The remaining amount of hydroxyl radicals could lead to the formation of aliphatic radicals. 

Figure 18A shows the excitation spectra recorded for the emission bands at 410 or 450 nm for an HSA solution containing 0.1 M LiBr before and after irradiation with a dose of 420 Gy. In the case of recording the excitation spectrum for the detection of 410 nm before irradiation, we observe two bands with a maximum of 292 nm and about 345 nm. For a detection length of 450 nm, we have three peaks in the excitation spectrum with a maximum at 292, 356 and 390 nm.

After irradiation, the same number of bands of excitation spectra is observed, but the intensity of the bands centered at about 330 nm increases significantly. This is especially visible for the detection wavelength of 410 nm, which is typical for DT. The analogous excitation spectra were obtained for the solution containing NaN_3_ (Figure 18B). The reaction of the $N_{3}^{•}$ radical with HSA results in similar spectral changes to those described for the ${\mathrm{Br}}_{2}^{•-}$ radical, but the DT band is less intense. These experiments clearly show that ${\mathrm{Br}}_{2}^{•-}$ and $N_{3}^{•}\;$penetrate to the HSA interior. ${\mathrm{Br}}_{2}^{•-}$ in particular reaches the tyrosine residues of the protein with high efficiency. The reaction of ${\mathrm{Cl}}_{2}^{•-}$ radicals with HSA leads to the formation of DT in a small amount (Figure 19A); however, in the reaction of albumin with ^•^OH radicals (Figure 19B), the formation of tyrosine–tyrosine bridges was not observed. It is more precise to say that we do not rule out bridging DT but rather bridging by such dityrosine isomers that may not be fluorescent [61]. An interesting observation is also the change in the amount of HSA intermolecular self-aggregates after the irradiation of N_2_O saturated solutions containing HSA (70 µM) and 0.1 M NaN_3_ or 0.1 M LiBr. In both cases, the number of light-emitting aggregates in the low-energy part of the band (an emission band extending over 375 nm, maximum band at ≈3.1 eV) decreased. This seems to be forced by the structure change of HSA dimers containing the DT covalent bond. The obtained results indicate that the DT formation efficiency does not correlate with the reactivity of the oxidizing radicals.

The oxidation process of HSA was monitored by pulse radiolysis measurements at 410 and 510 nm at the maximum of the absorbance band for TyrO^•^ and Trp^•^, respectively. Pulse radiolysis measurements show a correlation between the amount of the generated TyrO^•^ radicals (measured by the absorbance at 410 nm) and the fluorescence intensity DT recorded in steady-state measurements. However, the differences in the concentrations of TyrO^•^ radicals were not large. The highest absorbance value of the TyrO^•^ was recorded for the ${\mathrm{Br}}_{2}^{•-}$. The analysis of pulse radiolysis spectra for the TyrO^•^ at 410 nm is difficult due to broad absorption bands of the dihalide anion radicals in the spectral range at about 400 nm. The transient absorption spectra clearly revealed that for the reaction of ${\mathrm{Cl}}_{2}^{•-}$ and ^•^OH with albumin, less TyrO^•^ and Trp^•^ are formed in compared to other radicals. By analyzing the rate constants of ^•^OH and $N_{3}^{•}$ radicals with amino acids in a homogeneous solution, one can explain the reason for the significantly higher DT yield for the $N_{3}^{•}$ reaction with HSA in relation to the ^•^OH reaction with HSA. The publication [62] collects the values of the rate constants of $N_{3}^{•}$ and ^•^OH radicals with tryptophan, tyrosine, phenylalanine, histidine, cysteine, methionine and valine. The work shows that in the reaction with amino acids, the ^•^OH radical is more reactive than $N_{3}^{•}$ (which results from the difference in redox potential), but the difference is especially large in the case of non-aromatic amino acids (it is two or even three orders of magnitude). Of course, the ^•^OH radical can react with all the amino acids in HSA, but $N_{3}^{•}$ reacts mostly with Tyr and Trp. Our own study confirms that hydroxyl radicals are not effective in the formation of DT in histone solutions, and this is in line with previous research by others [63,64]. It can therefore be assumed that as a rule, the high reactivity of ^•^OH radicals with protein amino acids results in a lower relative oxidation efficiency of aromatic protein residues and a lower probability of forming dityrosine bridges. In the case of N_2_O-saturated and irradiated HSA solution, practically, we did not observe DT. It is probably related to the decay of TyrO^•^ radicals on the HSA surface. In order to induce DT formation, the TyrO^•^ radicals from different, modified protein molecules must react with each other. For non-selective, very reactive species, such as hydroxyl radicals, there is a high probability that the TyrO^•^ radical from one HSA molecule will react with a radical other than TyrO^•^ generated in another radiation-modified albumin molecule. The use of mild oxidizing species leads to the formation of only radicals centered on aromatic amino acid residues, mainly on or nearly the surface of proteins. In this case, the probability of recombination of TyrO^•^ radicals generated on the surfaces of two different HSA molecules increases, which leads to the formation of intermolecular DT. This does not exclude the participation of radicals generated within the HSA (e.g., at a tryptophan residue) in the formation of DT on the protein surface (or near the surface of the HSA where some Tyr residues are found).

It has long been known that within polypeptides, enzymes or proteins exposed to ionizing radiation, long-range electron transfer from tyrosine residues to semi-oxidized tryptophan moiety occurs. It can therefore be assumed that in the case of HSA, the residue of Trp214 will be involved indirectly in the formation of DT bridges. Such a hypothesis is confirmed by the observation of long-range electron transfer in HSA Tyr residue to the neutral tryptophan radical generated in pulse radiolysis [65]. This mechanism was proposed by Hashimoto et al. [59] to explain DT formation in lysozyme solutions as a result of the reaction of this protein with the radicals ${\mathrm{Br}}_{2}^{•-}$. Of course, due to the presence of only one tryptophan group in the structure of HSA, this reaction channel may be of less importance in our case.

It is interesting to compare changes in the emission spectra of HSA solutions caused by the reaction of hydroxyl or azide radicals with the tested protein used in high concentration. For this purpose, four HSA solutions with a concentration of 300 µM were studied, all saturated with N_2_O, and two additionally containing 0.1 M NaN_3_. The irradiated solutions (two out of four) absorbed a dose of 27.8 kGy. The excitation of the solutions with light with a wavelength of 320 nm leads to a wide emission band of HSA concentration (self-aggregates) aggregates (λ_max_ = 415 nm). The intensity of this band in the presence of azide is slightly lower, which is the result of the quenching of the excited states of aggregates by NaN_3_. This applies to all used excitation light wavelengths (range 295–410 nm). This effect has already been described by us. In the case of the reaction of $N_{3}^{•}$ radicals with HSA, an intermolecular DT bridge is generated, as evidenced by an intense fluorescence band with a maximum intensity at about 403 nm (see Figure 20).

The reaction of the ^•^OH radical with HSA leads mainly to species other than DT, as evidenced by a broad, less intense red-shifted band (λ_max_ = 414 nm, Figure 20). These species are the result of a recombination of radicals generated in a non-specific manner as a result of the recombination of radicals generated on the surface of HSA molecules. This leads to the formation of numerous covalent bridges (including C-C, it is also possible to generate C-S bonds), and consequently HSA dimers and aggregates. The resulting HSA nanostructures lead to light emission similar to that typical for concentration aggregates. This is clearly illustrated by Figure 21 summarizing the emission spectra recorded after excitation with light with a wavelength of 370 nm. The use of the correction procedure for the emission spectra (λ_exc_ = 320 nm) for the inner filter effect for HSA solutions (300 μM) exposed to the $N_{3}^{•}$ or ^•^OH radicals increases the intensity of the emission band by about five and four times, respectively, compared with the spectra without correction (Appendix A in Appendix A. In the case of applying the correction procedure of the emission spectrum resulting from the reaction of the ^•^OH radical with HSA, the shift of the band maximum from 414 nm (before correction) to 407 nm (after correction) is visible.

For the HSA solution, in which the ^•^OH radical reacted with the protein, the emission band significantly increases and differs in intensity from the remaining “concentration” bands, which are practically unchanged. This indicates that DT bridging does not increase the number of concentration aggregates (the high-energy ones, related to an excitation band with a maximum of about 350 nm; maximum band at 3.6 eV). An additional confirmation of this is the lower scattering of the excitation light by the solution irradiated in the presence of NaN_3_ as compared to the one in which the ^•^OH radical is generated, as shown in Figure 21 for the range 285–392 nm (gray circle). Of course, this is also observed in the excitation spectrum recorded at 475 nm. DT bridging is accompanied by an increase in the number of low-energy HSA self-aggregates emitting light above 500 nm; the maximum band at 3.1 eV (see Appendix A for λ_det_ = 525 nm in Appendix A). There are much fewer of them than those initiated by ^•^OH radicals. 

Excitation of the aqueous HSA solution (300 µM) with 390 nm light leads to the emission band centered at 455 nm. After irradiating this HSA solution saturated with N_2_O, the emission band was recorded with the maximum at 445 nm. The intensity of this band is 2.9 times higher with respect to the intensity of the emission spectrum of neat solution (see Appendix A in in Appendix A). The reaction of $N_{3}^{•}$ with HSA resulted in an emission band with a maximum of 470 nm (λ_exc_ = 390 nm). The intensity of this band is similar to the intensity of emission of neat solution. The use of the inner filter effect compensation procedure leads to an approx. 2-fold increase in the intensity of emission bands in irradiated solutions.

The HSA solution was analyzed by SDS-PAGE electrophoresis before and after irradiation with electron pulses (27.8 kGy) to detect the presence of monomer, dimer and aggregate molecules. First, 20 µL samples containing 30, 70, 300, and 600 µM HSA were tested before irradiation and N_2_O-saturated solution containing 300 µM HSA or 0.1 M NaN_3_ was tested after irradiation with a dose of 27.8 kGy. The electrophoresis results of non-irradiated solutions of HSA were consistent with emission measurements and showed a concentration-dependent aggregation. The aggregation of a commercial HSA was investigated by two different techniques, high-performance liquid chromatography and gel electrophoresis [66]. The higher the human serum albumin concentration, the more the monomer proportion decreases. Within the range of HSA concentrations of 58–435 µM, the dimer content in relation to the remaining protein fractions is about 24%. Our results, although not as precise as in [66], indicate a similar dimer contribution to other forms of HSA. The dimer is the most abundant object next to the monomer in HSA solutions in a very wide range of concentrations. It is important that in the conclusion from work [66], the HSA shows reversible aggregation with increasing concentration, which has been confirmed by us in spectroscopic measurements as a result of dilution of concentrated protein solutions.

It was evident that HSA aggregates increase in size after irradiation, regardless of the type of oxidant. The SDS-PAGE date indicates that the irradiation of HSA solution leads to the generation of aggregates with high MW (these aggregates are not able to enter in the running gel). 

After the irradiation of the solutions, the absorbance of the tested solutions increases. When HSA aggregates are formed, the increase in absorbance consists of two components of actual absorbance and “absorbance” due to scattering of the analyzing light. The correction is especially recommended for concentrated HSA solutions (above 300 µM). An example of the emission spectrum after correction for the inner filter effect is shown in Appendix A. To illustrate the effect of emission correction on the intensity and position of the emission spectrum, uncorrected and corrected spectra of HSA after oxidation by ^•^OH and $N_{3}^{•}$ radicals are compared (λ_exc_ = 390 nm). These experiments clearly revealed that after emission correction, the relative intensity of the HSA emission band in both (^•^OH and $N_{3}^{•}$) cases increases. In the case of HSA aggregates, which emits deeper into the red after excitation of the albumin solution with 390 nm light, the use of emission correction shifts the spectrum by 15 nm (445 nm→460 nm, ΔE = 0.09 eV) and 25 nm (470 nm→495 nm, ΔE = 0.0566 eV) for ^•^OH and $N_{3}^{•}$ radicals, respectively. 

In this study, we focused on the qualitative characteristics of HSA dimers and aggregates; therefore, it is not always necessary to correct the emission intensity values. Moreover, the absorption/emission spectra of albumin solutions with concentrations above 300 µM were verified by measuring the spectra of these solutions in a cuvette with an optical path length of 0.2 mm. We have found that 2 × 10 mm fluorescent cuvettes (excitation along the 2 mm optical path) significantly reduce the internal filter effect.

The scavenging of hydroxyl and azide radicals by protein is a complex process. The HSA consists of a single polypeptide chain of 585 amino acids. The free radicals, namely ^•^OH and $N_{3}^{•}$, may react with amino acids residues within albumin. We conducted a series pulse radiolysis experiment with aqueous solutions containing amino acids, including L-tyrosine, L-tryptophan, phenylalanine, methionine, histidine and tyrosine-tryptophan. Figure 22 presents the transient absorption spectra of primary products of HSA one-electron oxidation by azide radical (N_2_O-saturated 30 µM HSA solution containing 0.1 M NaN_3_). These spectra are compared with the spectra obtained after the oxidation of various amino acids in a series of pulse radiolysis experiments, each with one selected amino acid at a concentration of 1 mM. The measurements were carried out under anaerobic conditions to avoid subsequent reactions of the generated radicals with oxygen. ΔA denotes differentia absorbance with respect to that of the non-irradiated sample (before the pulse). We can distinguish three spectral regions in the transient absorption spectra of HSA: with maximum at 300, 410 and 520 nm. We are able to assign the aforementioned absorption bands to the products obtained as a result of the oxidation of Tyr and Trp within albumin by azide radicals: the band of the Tyr meta isomer centered at 300 nm and phenoxyl radicals TyrO^•^ with a maximum near 410 nm. The band with a maximum at 520 nm is attributed to the oxidation of the tryptophan molecule, ${\mathrm{Trp}}^{•}$:(6)$$\mathrm{TrpH}+{\;N}_{3}^{•}\;\to {\;\mathrm{TrpH}}^{•+}+{\;N}_{3}^{-},$$
(7)$${\mathrm{TrpH}}^{•+}\;↔{\;\mathrm{Trp}}^{•}+{\;H}^{+},$$

The reaction of $N_{3}^{•}$ with histidine and methionine in neutral solution is much slower than with other amino acids. The reaction between azide radicals and His or Met molecules is not observed.

When BSA reacts with $N_{3}^{•}$ radicals, similar transient absorption spectra were obtained as in the case of the reaction of azide radical with HSA. A significant difference was observed in the 450–570 nm region. The higher absorbance in the transition spectra for BSA after one-electron oxidation, observed above 500 nm (Appendix A in Appendix A), results from the presence of two tryptophan residues in the BSA structure and not one as for HSA. For BSA, the absorption band associated with Trp is about twice as intense as the tryptophan radical in HSA. It should be noted that the transient absorption in the 400 nm region (TyrO^•^) for the $N_{3}^{•}$ radical-induced oxidation of the HSA and BSA is the same.

The comparison of the absorption spectra recorded during the pulse radiolysis of aqueous solutions of several amino acids with analogous spectrum for HSA indicate that the mechanism of protein oxidation by hydroxyl and azide radicals differs significantly. Figure 23 shows the transient absorption spectra recorded after the electron-pulsed irradiation of N_2_O-saturated buffer solutions containing various amino acids or human serum albumin from a concentration of 30 µM. The transient absorption spectrum of HSA solution under oxidative conditions shows a maximum at 290 nm and in the spectral range 370–480 nm. The band with a maximum at 290 nm is due to products obtained as a result of the oxidation of Trp, Tyr, Phe, and His by hydroxyl radical. A broad absorption in the spectral range 370–480 nm can be attributed to the absorption of several oxidation products: Trp, Met, and His (superposition of absorption spectra of various amino acid oxidation products). The literature data indicate that methionine is oxidized to sulfoxide. However, this reaction is of minor importance in the process of peptide/protein oxidation in a N_2_O-saturated solution. The lack of a band with the maximum at approx. 510 nm attributed to the Trp^•^ in the transient absorption spectrum of HSA may indicate that hydroxyl radicals do not react with Trp214 within protein. It is less likely that the OH radical will reach Trp 214 and will react but not oxidize the tryptophan moiety. Referring to our earlier work [54] in which we showed that the hydrated electron does not reach the Sudlow 1 site, it can be concluded that this densely packed and hydrophobic domain is not directly exposed to radical attack. We noticed that which is assigned to the TyrO^•^ radical is formed in a longer time scale after the OH radical reaction with HSA (TyrO^•^ band is visible in the HSA spectrum after 170 µs after the electron pulse; see Appendix A). It can be assumed that this is the result of the transformation of the OH-adduct hydroxycyclohexadienyl radical) in the ortho position to the OH group into the phenoxyl radical of tyrosine: TyrO^•^. The conversion described above for the reaction of free tyrosine with the OH radical in the protein is slower. The amount of the TyrO^•^ radical generated in HSA is small, but it indicates that DT bridging in vivo as a result of OH radical attack on albumin is highly likely.

The literature describes that more than 94% of hydroxyl radicals form adducts on the aromatic ring of phenylalanine as a result of scavenging of ^•^OH [67]. The reaction of phenylalanine with a hydroxyl radical leads to the formation of 4-, 3-, and 2-hydroxyl-phenylalanine (para-, meta-, and ortho-tyrosine, respectively) and other Phe derivatives. Solar has proposed the preference position of ^•^OH addition to the phenylalanine ring: ortho > para > meta. Appendix A in Appendix A shows transient absorption spectra recorded after electron pulse irradiation of N_2_O-saturated buffer solution containing 1 mM phenylalanine. We can distinguish three spectral regions: broad absorption bands with maxima near 250, 270 and 320 nm. These bands were observed immediately after the irradiation of N_2_O-saturated Phe solution. The reaction of ^•^OH radicals with phenylalanine mainly forms an OH adduct in the ortho position. The absorption centered near 270 nm is probably due to 3,4-dihydroxyphenylalanine (DOPA). In the transient absorption spectrum of an HSA solution saturated with N_2_O exposed to an electron beam, it is difficult to recognize the bands derived from phenylalanine-OH adducts (Appendix A). This is due to the fact that other components of the protein attacked by the hydroxyl radical absorb light in this spectral range.

In our opinion, the absorption spectrum of the protein after the reaction with the ^•^OH radicals was more difficult to analyze due to the lack of selectivity of the hydroxyl radical. The ^•^OH radicals can react with protein molecules in three different ways (preferentially on the surface of albumin molecules) in contrast to a specific one-electron oxidant, the azide radical. Taking into account the rate constants of the $N_{3}^{•}$ and ^•^OH radicals’ reaction with individual amino acids, it is safe to postulate that the absorption spectrum of the oxidized protein does not have to correspond to the absorption spectra of the isolated oxidized amino acids.

## 3. Discussion

In this article, we examined the processes accompanying the formation of HSA aggregates that are formed in irradiated protein solutions as a result of the attack of oxidizing radicals. In these studies, gamma rays and electron beam irradiation were used to treat HSA solutions in a very wide concentration range from 10 µM to 1 mM. For the analysis of the obtained nanostructures, we used spectroscopic techniques, including the analysis of absorption and emission spectra and the measurement of the lifetime of the excited states of both monomers and the resulting HSA aggregates. As methods complementing the analysis, we used dynamic light scattering and gel electrophoresis measurements. The measurements are not straightforward and require considerable effort due to the spontaneous formation of HSA aggregates prior to irradiation. We confirmed that not only HSA but many other proteins and enzymes (the results for bovine serum albumin, ovalbumin, and papaine will be presented soon in a subsequent publication) undergo concentration-dependent reversible aggregation as a result of protein–protein interactions (hydrogen and/or ionic bonds are formed between protein molecules). There are two populations of self-aggregates in a solution of human serum albumin, even at a low concentration of HSA. The consequence of the formation of HSA self-aggregates is the appearance of a new “blue” emission in the spectrum of the albumin solution.

The light emission of protein aggregates was not related to the presence of aromatic amino acids in the peptide chain and depended almost linearly on the concentration of HSA. Our measurements do not support the thesis that there is a concentration threshold (around 10 µM) above which self-aggregates of HSA are formed. In our opinion, dimers (two HSA molecules linked together by a non-covalent bond) are mainly responsible for blue emissions. One should also consider the formation of different dimer types (depending on the structural changes of the protein). In this study, we focused mostly on the qualitative characterization of the HSA dimers and aggregates. It is due to the unknown value of the emission quantum yield of HSA aggregates. The time-resolved fluorescence measurements for HSA solution revealed that two populations of protein aggregates are presented in the HSA solution. Furthermore, our experiments confirmed that temperature and pH influence the aggregation process of HSA. The influence of oxygen on the HSA aggregation process is insignificant.

The formation of concentration-dependent dimers/aggregates of albumin is a reversible process (dilution of the albumin solution leads to a disaggregation of the HSA). The HSA aggregates are not strongly modified under the influence of ionic surfactants (20 and 60 mM SDS and 60 mM DTAC) and solvents such as *t*-BuOH: PBS 1: 1, D_2_O, EG: H_2_O 1:1, ethylene glycol: water 1:1 matrix in 77 K. Freezing the HSA solution allowed the separation of two populations of aggregates. Increasing the temperature and lowering the pH of the solution promoted the formation of HSA aggregates. As the temperature rises, the protein solvation layer is destroyed, and the aggregation process takes place (conducted experiments suggest the presence of “dynamic” aggregates). Thermally produced aggregates differ in structure from concentration-dependent aggregates. Our experiments established that the protein aggregation process was modified by the presence of salt in the solutions (2 M or 7 M NaClO_4_, 2 M NaSCN and 2 M NaI). In addition, we found that NaClO_4_ significantly changes the structure of water, while NaSCN and NaI have little effect on its structure. In the case of HSA solutions with a concentration lower than the physiological concentration of albumin (600 μM), no changes in the water structure caused by the presence of HSA were observed. Significant changes in the water structure were observed in the solution containing HSA with a concentration above 760 μM.

In the measurements of radiolysis, we made a comprehensive comparison of the reactions of oxidizing radicals with albumin and these radicals with selected HSA components (e.g., tryptophan, tyrosine, phenylalanine, cystine). Since the main goal of our publication is to study protein aggregation through spectroscopic analysis of the process of creating intermolecular bridges of HSA tyrosine residues, we started our measurements with a radiolysis of aqueous tyrosine solutions. The DT formation process in the buffer solution depends on the pH of the solution and the type of ionizing radiation (electron beam or γ radiation). The dependence of the DT emission intensity as a function of the dose absorbed by Tyr solutions at different pH values was linear. Dityrosine can be used as a marker of protein aggregates formed under oxidative conditions in solutions with a low albumin concentration (a low albumin concentration allows elimination/minimization emissions from self-aggregates). As a result of the irradiation of protein solutions during oxidative stress, the generation of DT is manifested by an emission band with a maximum of about 400 nm after the excitation of irradiated solution with the 315–325 nm light. The process of DT formation was also analyzed in irradiated tyrosine solution in the presence of amino acids. Regardless of the type of amino acids added to the tyrosine solutions (aromatic, sulfur-containing or other amino acids within the HSA), the irradiation of these solutions in the presence of oxidizing radicals leads to typical DT fluorescence. The reaction of amino acids with azide radicals causes the generation of higher amounts of DT compared to reactions with hydroxyl radicals.

An excess of dimeric and oligomeric forms of albumin may be the first symptom of neoplastic or neurodegenerative diseases (e.g., Alzheimer’s disease). For this reason, we became interested in spectroscopic techniques for the identification of DT and protein aggregates with the detection of light emission. The irradiation of HSA solutions under oxidative stress conditions (as a result of the reaction of ^•^OH, ${\mathrm{Br}}_{2}^{•-}$, ${\mathrm{Cl}}_{2}^{•-}$, $N_{3}^{•}$, ${\mathrm{SO}}_{4}^{•-}$ or $N_{3}^{•}$ radicals) results in the formation of stable protein aggregates with low and high molecular weight. Aggregates produced by irradiation are irreversible. These two types of aggregates (for small doses of ionizing radiation, mainly dimers are formed) lead to the formation of covalent bonds. The conducted experiments confirmed that the mechanism of the ${\mathrm{Cl}}_{2}^{•-}$ and hydroxyl radical reactions with albumin differs significantly with respect to reactions of HSA with mild oxidants (${\mathrm{Br}}_{2}^{•-}$, $N_{3}^{•}$, ${\mathrm{SO}}_{4}^{•-}$). In the latter case, radiation-induced aggregates show specific emission different from non-irradiated solutions. The emission intensity of HSA aggregates formed as a result of the reaction of albumin with ^•^OH radicals depends linearly on the absorbed dose (regardless of the excitation wavelength). In the case of azide radicals, the intensity of emission of HSA aggregates increases monotonically but non-linearly. These aggregates are characterized by classic DT fluorescence. Analysis of the oxidation process of HSA (70 μM) with the use of a number of oxidants generated by radiation showed that as a result of oxidation of human serum albumin with mild oxidants (${\mathrm{Br}}_{2}^{•-}$, $N_{3}^{•}$, ${\mathrm{SO}}_{4}^{•-}$), covalent bonds between tyrosine residues are formed. It is manifested by an intense emission band with a maximum of about 400 nm after the excitation of irradiated solution with the 320 nm light. In this case of ${\mathrm{SO}}_{4}^{•-}$ radicals, we observed the superposition of fluorescence of dityrosine and the emission of HSA aggregates. The efficiency of DT formation does not correlate with the reactivity of the oxidizing radicals. Our measurements clearly revealed that ${\mathrm{Br}}_{2}^{•-}$ and $N_{3}^{•}\;$penetrate to the HSA interior, and the bromide anion radical in particular reaches the tyrosine residues of the protein with high efficiency. Both the redox potentials and rate constants of the studied oxidizing species with amino acids allowed determining the sequence of their reactivity: ^•^OH (2.7 V vs. SHE) > ${\mathrm{SO}}_{4}^{•-}$ (2.43 V vs. SHE) > ${\mathrm{Cl}}_{2}^{•-}$ (2.1 V vs. SHE) > ${\mathrm{Br}}_{2}^{•-}$ (1.7 V vs. SHE) > $N_{3}^{•}$ (1.23 V vs. SHE) [68]. The efficiency of DT formation between HSA molecules induced by studied radicals did not correlate with their reactivity: ${\mathrm{Br}}_{2}^{•-}\;$ > ${\mathrm{SO}}_{4}^{•-}$ > $N_{3}^{•}\;$ > ${\mathrm{Cl}}_{2}^{•-}$ > ^•^OH. The lack of dependence between the redox potential of oxidizing radicals and the efficiency of DT formation may result from the high reactivity and, at the same time, lower selectivity of ^•^OH and ${\mathrm{Cl}}_{2}^{•-}\;$ radicals compared to ${\mathrm{Br}}_{2}^{•-}\;$ and $N_{3}^{•}$. The ^•^OH and ${\mathrm{Cl}}_{2}^{•-}$ radicals attack randomly available places on the albumin surface, leading to the formation of the α carbon-centered free radical of HSA [69,70]. These radicals are formed as a result of the abstraction of a hydrogen atom from the α-carbon by oxidizing radicals, e.g., ^•^OH radicals.

This raises the question of the impact of HSA concentration on the aggregation process. We performed steady-state radiolysis experiments for HSA solutions with a concentration of 300 µM, which are saturated with N_2_O and containing 0.1 M NaN_3_. The irradiated solutions absorbed a dose of 27.8 kGy. It was obvious that the type of generated intermolecular bonds depends on the used oxidizing radicals. As in the case of the 70 µM HSA solution, the reaction of $N_{3}^{•}$ radicals with HSA (HSA solution containing 300 µM) generates an intermolecular DT bridge, as evidenced by an intense fluorescence band with a maximum intensity at about 403 nm. In N_2_O-saturated and irradiated HSA solution, we did not observe typical emission from DT (λ_max_ = 403 nm) but a broad, less intense red-shifted band (λ_max_ = 414 nm). Hence, the reaction of the hydroxyl radical with HSA leads mainly to species other than DT and to the light emission typical of concentration-dependent aggregates. These new forms of HSA are the result of the recombination of radicals generated in a non-specific manner as a result of the recombination of radicals generated on the surface of HSA molecules. This leads to the formation of numerous covalent bridges (including C-C, it is also possible to generate C-S bonds) and consequently HSA dimers and high-energy aggregates (emission band at 3.6 eV).

We are aware that various types of tyrosine dimers can be formed in proteins after oxidative attack. More than five different DT isomers were identified in irradiated protein and peptide solutions and four are identified for the tyrosine solution [47,61]. Dityrosine is used in medicine as a biomarker for various diseases, including neurodegenerative diseases, but biological tests detect only one type of DT (ortho-ortho) dimer. Different dityrosine isomers may have different fluorescent properties (or may not be fluorescent). Therefore, their detection and separation in biological samples can be difficult. For this reason, the mechanism of tyrosine oxidation under various measurement conditions should be thoroughly investigated, which will be the basis for the development of a procedure for the detection of DT dimers in biological assays.

First of all, the measurements should start with recording the absorption and emission spectra of the analyzed protein solution in a wide range of concentrations (30–600 µM) before irradiation (preparation of a calibration curve). Emission experiments should be performed at different excitation wavelengths, not only at 310–320 nm (excitation wavelength DT). The emission of spontaneous HSA aggregates is a difficulty in the analysis of emission spectra related to radical processes occurring as a result of irradiation. Although formally, the intensity of fluorescence (emission) is not an additive quantity, the difficulties related to the emission of concentration aggregates can be corrected by manipulating the spectra (subtracting from the spectrum of the newly generated emission coming from the aggregates generated by radiation, the spectra recorded before irradiation) or by modifying the method of recording the spectra. The simplest method for the partial or even complete elimination of the inner filter effect is to use cuvettes with a short optical path or a triangular cuvette (for use in front-face illumination experiments). When the emission spectrum of DT bridges (which are the result of ionizing radiation) appears as an additional component of the spectrum of the non-irradiated HSA solution, the subtraction procedure is quite successful. This is due to the fact that the peak of the fluorescence spectrum DT is significantly shifted (Figure 17) in relation to the emission band peak of self-aggregates. As shown in the above-mentioned Figure 17, the emission spectra recorded after the reaction of the ^•^OH and ${\mathrm{Cl}}_{2}^{•-}$ radicals with HSA do not differ significantly in their intensity and shape from the spectra before irradiation. In this case, the analysis of the contribution of the newly produced aggregates to their total quantity by means of the emission detection method is very difficult. Recently, a technological platform based on radiation cross-linked proteins and enzymes has been developed to obtain materials of biomedical importance. In our upcoming publications, we intend to address to important mechanistic aspects of synthesis by irradiation protein and enzymatic aggregates in line with the latest technological findings [6,60,71,72,73,74].

## 4. Materials and Methods

### 4.1. Sample Preparation

Essentially fatty acid free albumin from human serum (HSA) and studied amino acids were obtained from Sigma-Aldrich (St. Louis, MO, USA)/Merck (Darmstadt, Germany) and was used as received. All solutions were prepared in ultrapure water in phosphate buffer 10 mM pH 7.2. Water was purified with the Hydrolab SPRING 20UV system. HSA was dissolved in PBS solution to appropriate concentrations immediately before measurements. Final concentrations of the solutions were verified spectrophotometrically using the molar absorption coefficients: _ε280 nm_ = 35,500 M^−1^ cm^−1^ for HSA [43], ε_275 nm_ = 1410 M^−1^ cm^−1^ for tyrosine, ε_280 nm_ = 5600 M^−1^ cm^−1^ for tryptophan, ε_257 nm_ = 20,700 M^−1^ cm^−1^ for methylviologen [75].

### 4.2. Optical Measurement

Steady-state emission and excitation spectra of examined solutions were carried out using Aminco-Bowman Series 2, equipped with a xenon lamp and a red-sensitive photomultiplier (Hamamatsu R928). Typically, the voltage of the photomultiplier divider was 650 V. The excitation wavelength was 295 nm for exciting Trp214 and above 320 nm for exciting HSA aggregates. The excitation and emission slits were set to 4.0 and 2.0 nm, respectively. Fluorescence and emission spectra presented in the publication are not corrected for the instrument’s response. Our spectrofluorimeter enables easy correction thanks to correction factor files, which are supplied with the software. Correction for the spectral response of the apparatus does not introduce any significant modifications to the spectra in the range of 200–600 nm (see Appendix A in Appendix A). We chose not to use this procedure for practical reasons. Comparing the so corrected spectra recorded for different spectrofluorimeter operating parameters (scanning speed, slits, etc.) does not significantly improve the data analysis. Certainly, the compensation of the inner filter effect is more important.

In the emission measurement, the inner filter effect refers to the decrease in the quantum efficiency of the emission and/or the deformation of the band shape due to the reabsorption of the emitted radiation. The fluorescence data can be corrected using the following described equation:I_corrected_ = I_recorded_·10^(Aexc+Aem)/2^,(8)
whereI_recorded_—observed emission intensity;
I_corrected_—corrected emission intensity;
A_exc_—absorbance of the solution at the excitation wavelength;
A_em_—absorbance of the solution at the emission wavelength.


The shape of the fluorescence spectrum recorded on different types of spectrofluorimeters under identical conditions may differ, because the sensitivity of photodetectors and the efficiency of some optical elements (e.g., monochromators) depend on the wavelength. When it is required to compare the fluorescence intensity at different wavelengths, the spectral calibration procedures must take into account the efficiency and sensitivity of the components involved in the construction of the spectrofluorimeter. This is usually completed using a sample with a known emission spectrum. In our case, we use quinine, tyrosine, and tryptophan solutions. The comparison of the tyrosine and tryptophan fluorescence spectra recorded with the AB2 spectrofluorimeter with the corrected emission spectra taken from the literature gave a very good agreement. The quinine spectrum obtained on the AB2 spectrofluorimeter is only slightly different from that in the literature. In order to compare the spectra of HSA aggregates recorded above 475 nm, we used the correction factor, taking into account the difference between our spectrum and the one in the literature [32]. 

Time-resolved fluorescence measurements were described elsewhere [76].

Absorption spectra were recorded with the resolution of 0.5 nm using a Perkin Elmer Lambda 750 spectrophotometer. Measurements were made using a 2 mm or 1 cm quartz cuvette. The monochromatic beam is partly transmitted, partly absorbed and partly scattered by the sample. Measurements performed on scattering samples such as irradiated HSA solution (above 30 µM) require a correction of the absorbance signal. In order to accurately analyze the absorption spectra of HSA solution, the absorbance signal was corrected to the smallest value of A for the highest value of the spectrum wavelength. In the case of concentrated protein solutions irradiated with high doses, the samples were irradiated in 1 cm quartz cuvettes, and the absorption and emission spectra were measured in the same cuvettes.

### 4.3. Pulse Radiolysis

Pulse radiolysis was performed using a 6 MeV linear accelerator (LINAC) ELU-6E operating in a single pulse mode. The equipment for pulse radiolysis with the optical detection has been described elsewhere [43]. The flow system was used in the pulse radiolysis of HSA and amino acid solutions with the volume of 250 mL. The solutions were N_2_O or N2 saturated before irradiation. Water radiolysis and the process of formation of reactive radical species used to initiate radical reactions have been described in detail in the literature [68]. In the pulse radiolysis measurements, we used two measurement regimes. In the nanosecond domain, we most often used a pulse with a duration of 17 ns generating 55 Gy. The dose could be varied over a wide range from 10 (pulse duration 7 ns) to 60 Gy. The dose was determined for each measurement series using water [54] or thiocyanate as a dosimeter. We also used two different pulses with a duration of 1 or 4 µs. The typical dose for a 1 µs pulse is 220 Gy, and for a 4 µs pulse, it is around 800 Gy. Here, we used an alanine dosimeter to measure the dose. For irradiation in steady-state measurements, we used a radiation chamber with the initial activity of Co-60 radiation sources 2200 TBq (60 kCi). The system was based on a panoramic gamma Ob-Servo-D radiator.

## Figures and Tables

**Figure 1 ijms-23-08090-f001:**
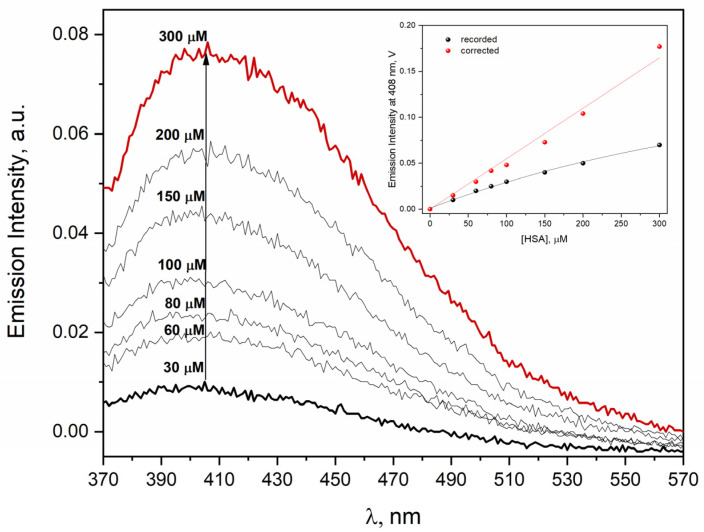
Emission spectra of the neat HSA solutions (30–300 µM). The excitation wavelength was 320 nm. **Insert:** Dependence of I_max_ as a function of HSA concentration before and after inner filter effect correction. I_max_—intensity of HSA emission in buffer solution at maximum of emission band, λ_exc_ = 320 nm. Emission intensity correction procedure is described in Materials and Methods.

**Figure 2 ijms-23-08090-f002:**
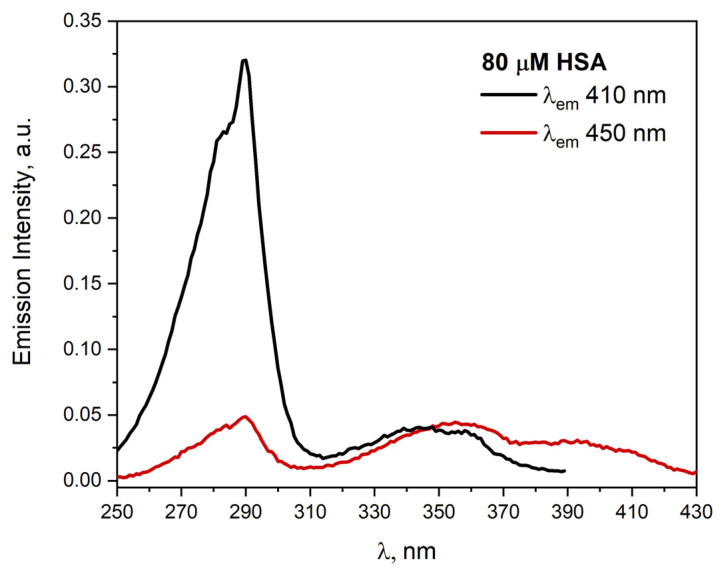
Emission excitation spectra of the neat HSA solution (80 µM). The emission wavelengths were 410 and 450 nm.

**Figure 3 ijms-23-08090-f003:**
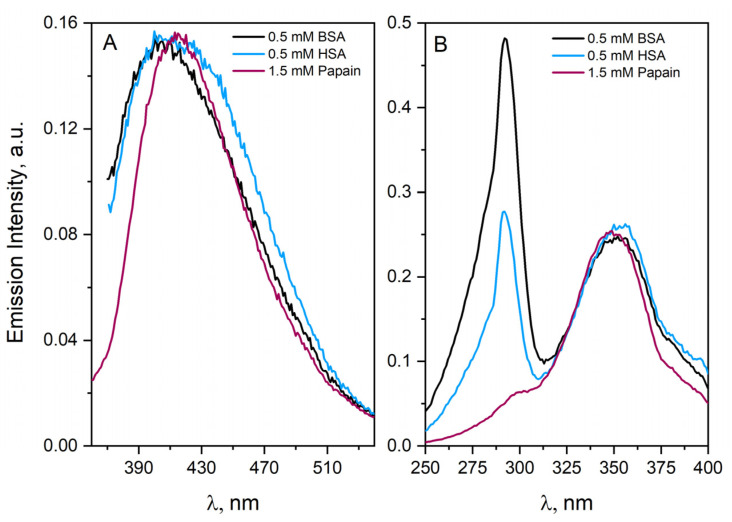
(**A**) Emission spectra of the neat HSA (0.5 mM), BSA (0.5 mM) and papain (1.5 mM) solutions. The excitation wavelength was 320 nm. Spectra were normalized. (**B**) Emission excitation spectra of the neat HSA (0.5 mM), BSA (0.5 mM) and papain (1.5 mM) solutions. The emission wavelength was 430 nm. Spectra were normalized.

**Figure 4 ijms-23-08090-f004:**
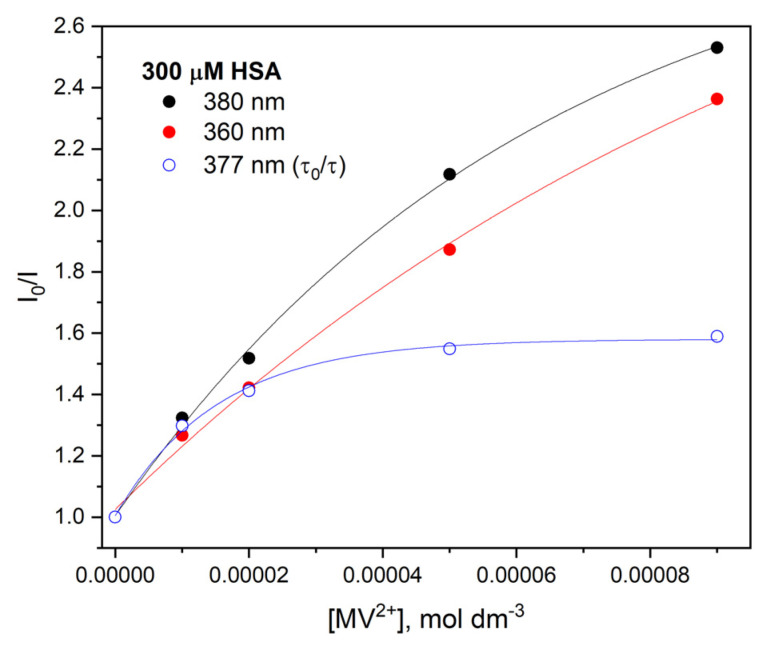
Stern–Volmer plot of changes in fluorescence intensity (I_0_/I) of HSA (300 µM) versus the concentration of MV^2+^ (0–90 mM). The I_0_ and I value for a given quencher concentration were read from the HSA emission spectra recorded in the absence and presence of MV^2+^, respectively (λ_max_ = 360 and 380 nm). The τ_0_ and τ value denote the fluorescence lifetimes of HSA in the absence and presence of MV^2+^, respectively (λ_exc_ = 337 nm).

**Figure 5 ijms-23-08090-f005:**
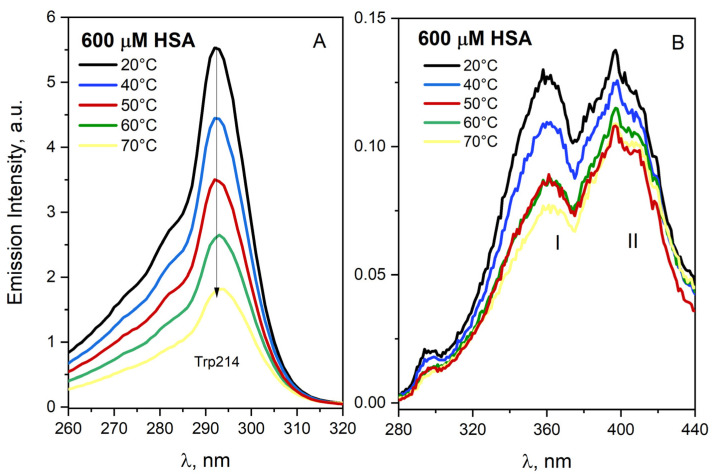
(**A**). Emission excitation spectra of the neat HSA solution (600 µM). The emission wavelength was 345 nm. (**B**). Emission excitation spectra of the HSA aggregates (600 µM). The emission wavelength was 500 nm. I and II—number of excitation bands.

**Figure 6 ijms-23-08090-f006:**
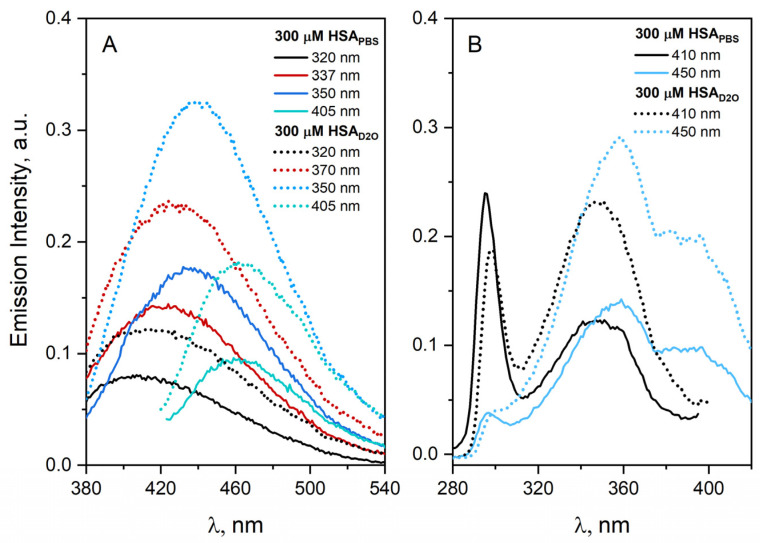
(**A**) Emission spectra of the neat HSA solution (300 μM) and HSA solution (300 μM) containing D_2_O. The excitation wavelengths are given in the figure. (**B**) Emission excitation spectra of the neat HSA solution (300 μM) and HSA solution (300 μM) containing D_2_O. The emission wavelengths are given in the figure.

**Figure 7 ijms-23-08090-f007:**
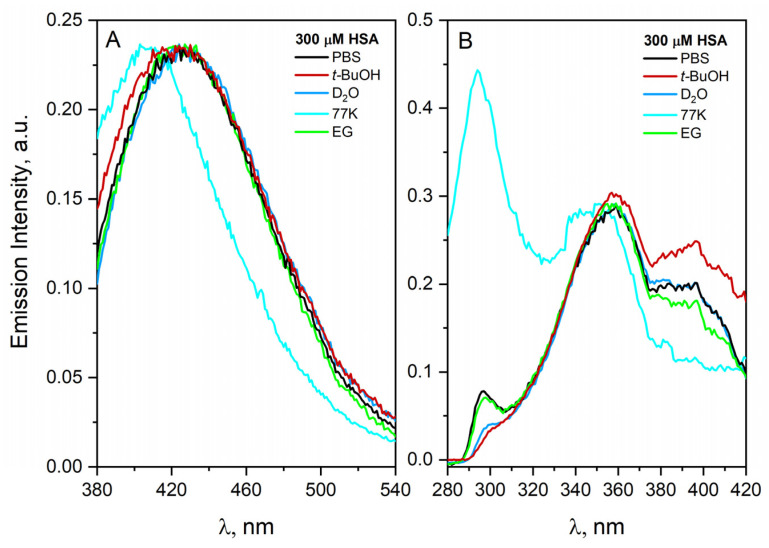
(**A**) Emission spectra of the HSA (300 μM) in various solvent: PBS, *t*-BuOH: PBS 1:1, D_2_O, EG:H_2_O 1:1 and EG:H_2_O 1:1 in 77 K. The excitation wavelength was 337 nm. (**B**). Emission excitation spectra of the HSA (300 μM) in various solvent: PBS, *t*-BuOH: PBS 1:1, D_2_O, EG:H_2_O 1:1 and EG:H_2_O 1:1 in 77 K. The emission wavelength was 450 nm. Spectra were normalized.

**Figure 8 ijms-23-08090-f008:**
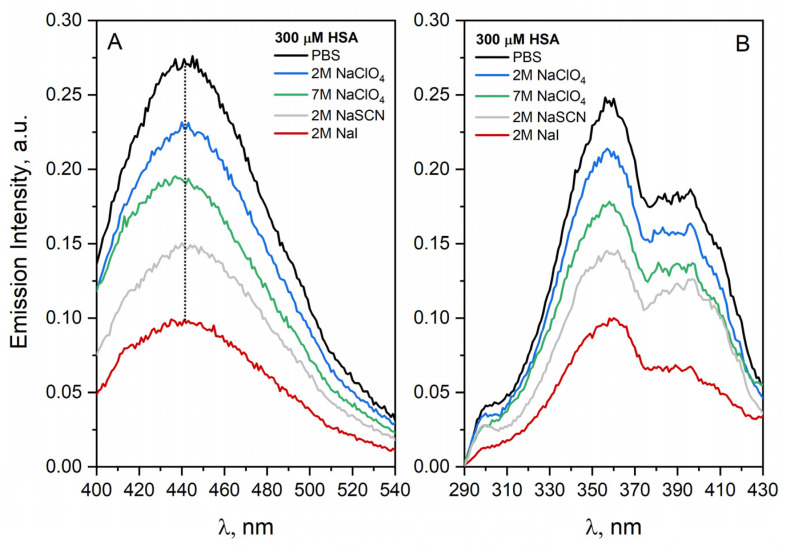
(**A**). Emission spectra of the HSA solutions (300 μM) containing NaClO_4_ (2M, 7M), NaSCN (2M) and NaI (2M). The excitation wavelength 360 nm. (**B**). Emission excitation spectra of the HSA solutions (300 μM) containing NaClO_4_ (2M, 7M), NaSCN (2M) and NaI (2M). The emission wavelength is 450 nm.

**Figure 9 ijms-23-08090-f009:**
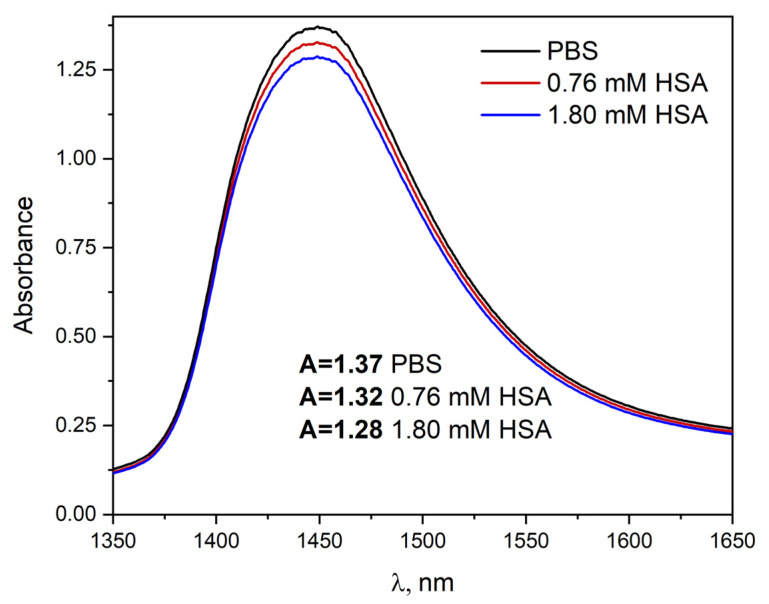
Absorption spectra of buffer solution of HSA (0.76 and 1.80 mM). Measurements were made using a 0.2 cm quartz cuvette. The PBS plot represents the spectra in a wide range of HSA concentration (15 to 600 µM).

**Figure 10 ijms-23-08090-f010:**
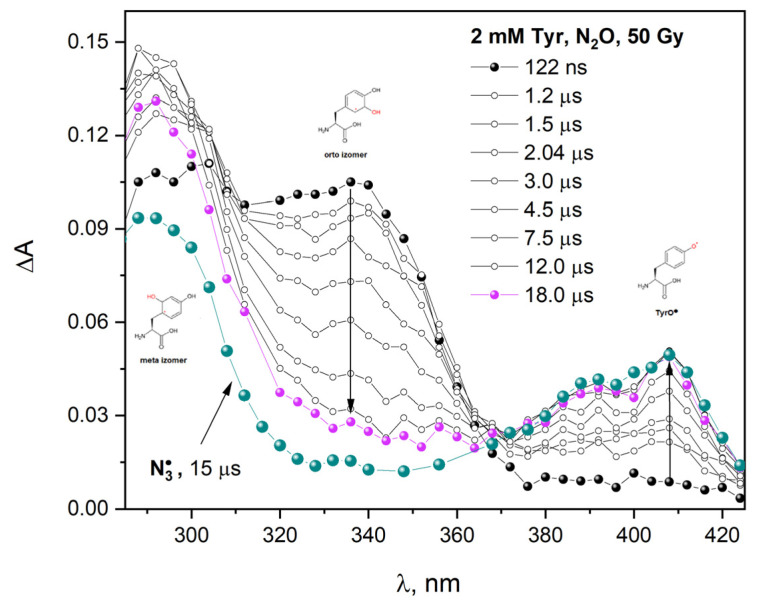
Transient absorption spectra of N_2_O-saturated buffer solution containing 2 mM Tyr, obtained for an irradiation dose of 60 Gy. Transient absorption spectra of N_2_O-saturated buffer solution containing 2 mM Tyr and 0.1 M NaN_3_, obtained for an irradiation dose of 60 Gy.

**Figure 11 ijms-23-08090-f011:**
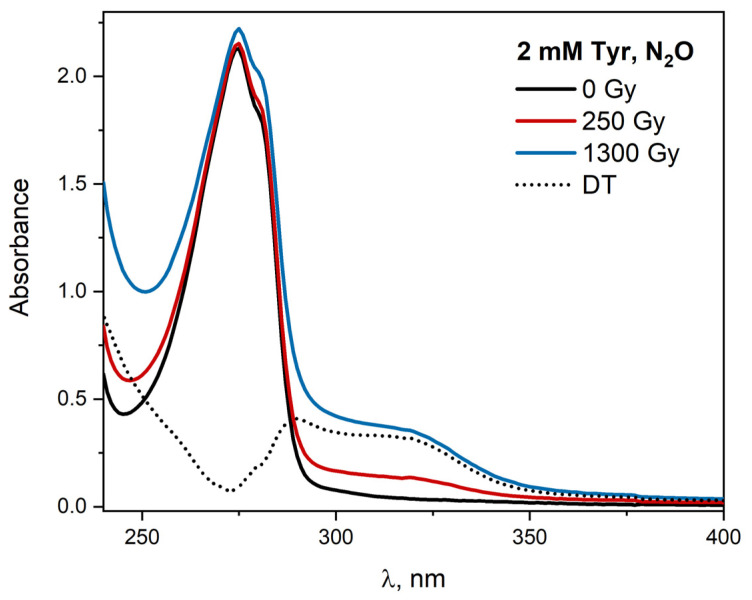
Absorption spectra of N_2_O-saturated buffer solution of Tyr (2 mM), obtained for irradiation doses between 0 and 1300 Gy.

**Figure 12 ijms-23-08090-f012:**
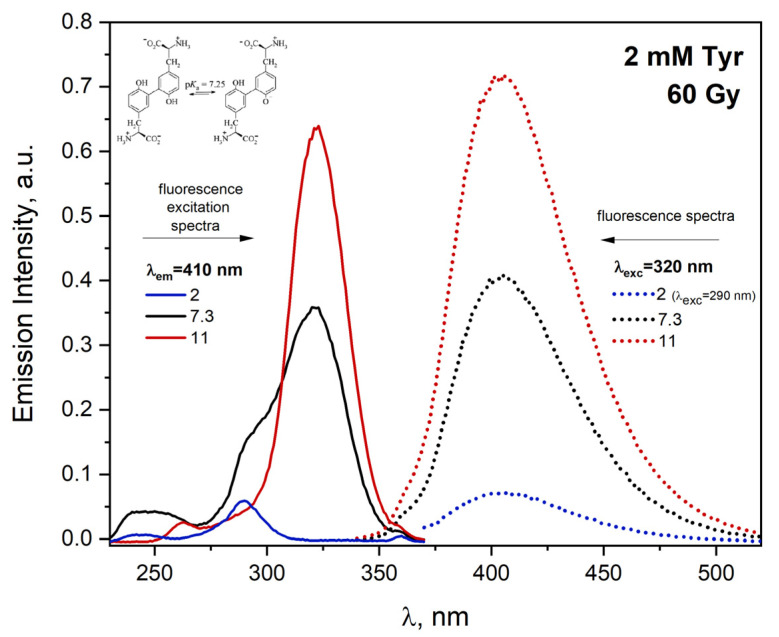
Emission spectra of the N_2_O-saturated Tyr solutions (2 mM) under pH: 2, 7.3 and 11, recorded after irradiation with a dose 60 Gy. The excitation wavelength was 320 nm. Emission excitation spectra of the N_2_O-saturated Tyr solutions (2 mM) under pH: 2, 7.3 and 11, recorded before and after irradiation with a dose 60 Gy. The emission wavelength was 410 nm.

**Figure 13 ijms-23-08090-f013:**
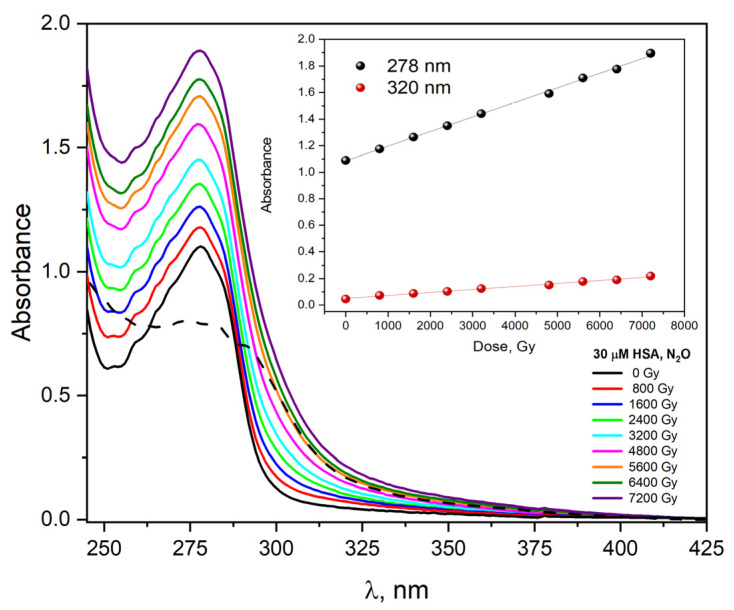
Absorbance spectra of N_2_O-saturated buffer solution of HSA (30 μM), obtained for irradiation doses between 0 and 7200 Gy. **Insert**: Dependence of A_278 nm_ and A_320 nm_ as a function of dose (0–7200 Gy).

**Figure 14 ijms-23-08090-f014:**
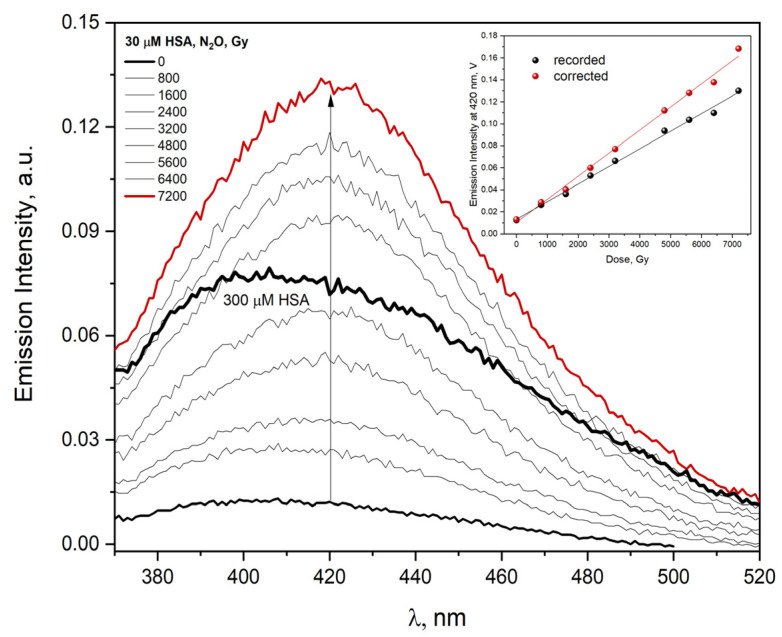
Emission spectra of N_2_O-saturated buffer solution of HSA (30 μM), obtained for irradiation doses between 0 and 7200 Gy. Emission spectrum of neat HSA solution (300 μM). The excitation wavelength was 320 nm. The arrow presented in the figure shows the increase in the intensity of the emission signal as a function of the absorbed dose. **Insert**: Dependence of I_max_ as a function of dose (0–7200 Gy) before and after inner filter effect correction.

**Figure 15 ijms-23-08090-f015:**
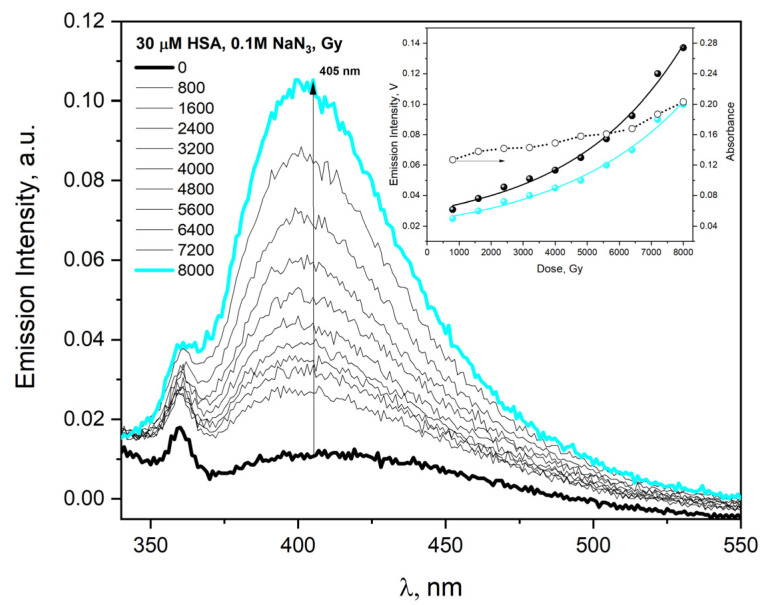
Emission spectra of N_2_O-saturated buffer solution of HSA (30 μM) containing NaN_3_ (0.1 M), obtained for irradiation doses between 0 and 8000 Gy. The excitation wavelength was 320 nm. The arrow presented in the figure shows the increase in the intensity of the emission signal as a function of the absorbed dose. **Insert**: Dependence of Imax as a function of dose (0–8000 Gy) before (blue circles) and after inner filter effect correction (black circles). Dependence of Amax as a function of dose (0–8000 Gy) (white circles).

**Figure 16 ijms-23-08090-f016:**
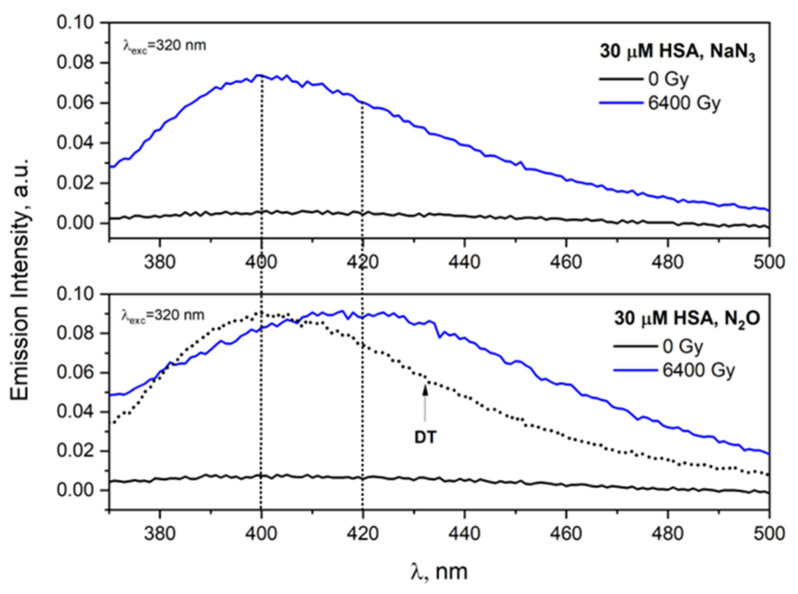
Emission spectra of N_2_O-saturated buffer solution of HSA (30 μM) containing NaN_3_ (0.1 M), recorded before and after irradiation with a dose 6400 Gy. Emission spectra of N_2_O-saturated buffer solution of HSA (30 μM), recorded before and after irradiation with a dose 6400 Gy. The excitation wavelength was 320 nm. Measurements were made using a 0.2 cm quartz cuvette. The dashed lines shows λ_max_ for DT (400 nm) and λ_max_ for aggregates formed in the reaction of HSA with ^•^OH radicals (429 nm).

**Figure 17 ijms-23-08090-f017:**
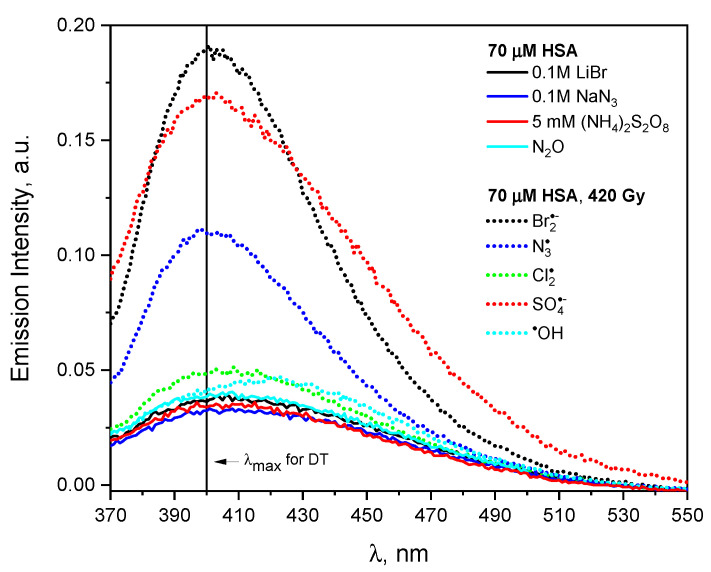
Emission spectra of the N_2_Osaturated HSA solutions (70 μM) containing 0.1 M LiBr, 0.1 M NaCl, 0.1 M NaN_3_, 5 mM (NH_4_)_2_S_2_O_8_ recorded before and after irradiation with a dose 420 Gy. *t*-BuOH was added to the HSA solution containing (NH_4_)_2_S_2_O_8_. The excitation wavelength was 320 nm. The vertical line shows λ_max_ for DT (400 nm).

**Figure 18 ijms-23-08090-f018:**
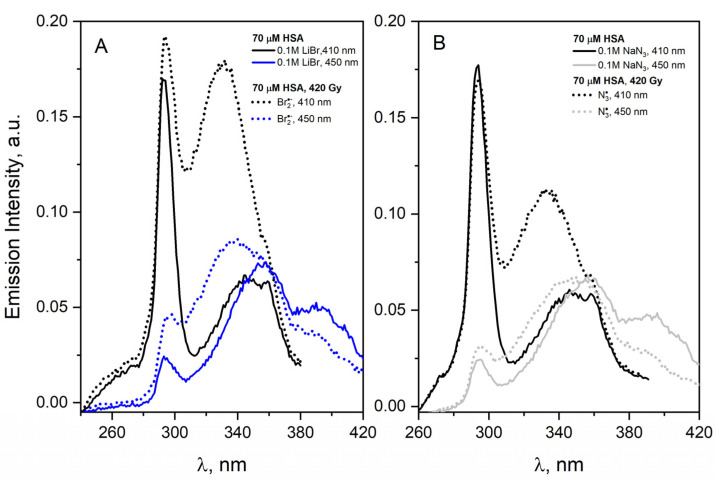
Emission excitation spectra of the N_2_O-saturated HSA solutions (70 μM) containing 0.1 M LiBr (**A**) or 0.1 M NaN_3_ (**B**), recorded before and after irradiation with a dose of 420 Gy. The emission detection wavelengths are 410 and 450 nm.

**Figure 19 ijms-23-08090-f019:**
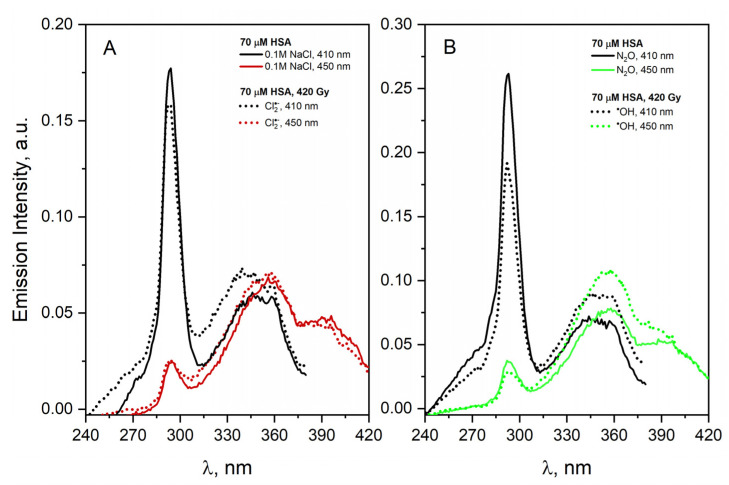
(**A**). Emission excitation spectra of the N_2_O-saturated HSA solutions (70 μM) containing 0.1 M NaCl recorded before and after irradiation with a dose 420 Gy. The emission detection wavelengths: 410 and 450 nm. (**B**). Emission excitation spectra of the N_2_O-saturated HSA solutions (70 μM) recorded before and after irradiation with a dose 420 Gy. The emission detection wavelengths: 410 and 450 nm.

**Figure 20 ijms-23-08090-f020:**
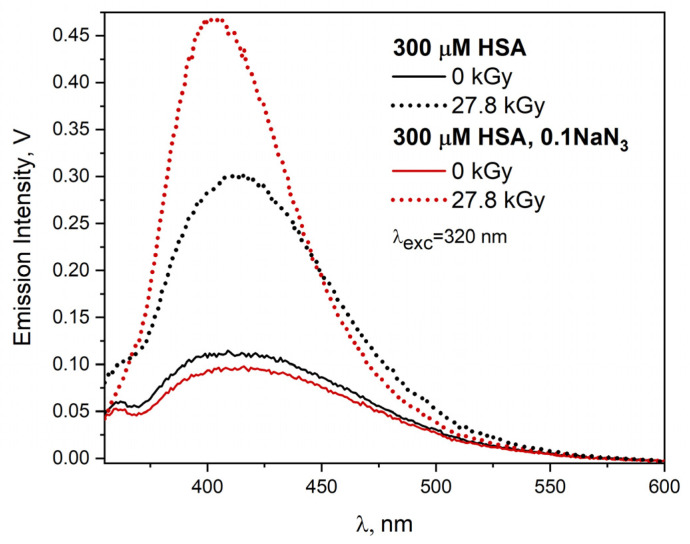
Emission spectrum of neat HSA solution (300 μM) and HSA solution containing NaN_3_ (0.1 M). Emission spectra of N_2_O-saturated buffer solution of HSA (300 μM) and containing NaN_3_ (0.1 M), obtained for irradiation dose 27.8 kGy. The excitation wavelength was 320 nm.

**Figure 21 ijms-23-08090-f021:**
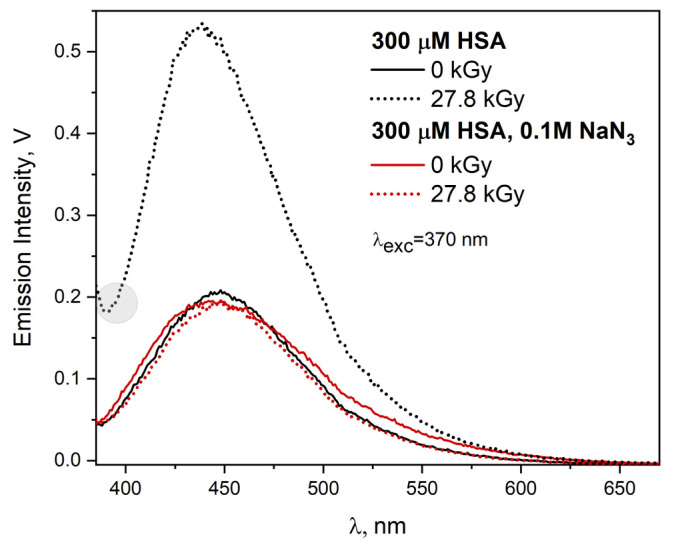
Emission spectrum of neat HSA solution (300 μM) and HSA solution containing NaN_3_ (0.1 M). Emission spectra of N_2_O-saturated buffer solution of HSA (300 μM) and containing NaN_3_ (0.1 M), obtained for irradiation dose 27.8 kGy. The excitation wavelength was 370 nm.

**Figure 22 ijms-23-08090-f022:**
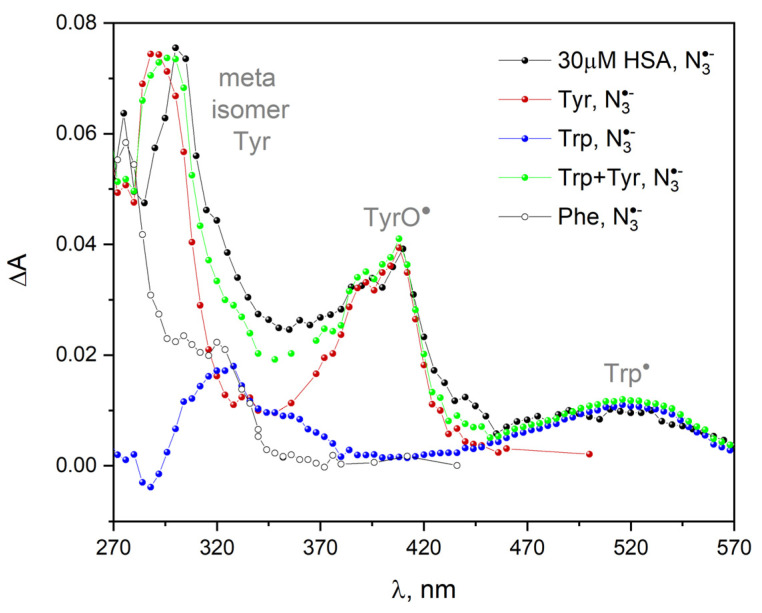
Transient absorption spectra of N_2_O-saturated buffer solutions containing: 30 μM HSA, 1 mM Tyr, 1 mM Trp, 1 mM Trp + Tyr, 1 mM Phe and 0.1 M NaN_3_, obtained for irradiation dose of 800 Gy. Each of the above-mentioned solutions was subjected to radiolysis separately.

**Figure 23 ijms-23-08090-f023:**
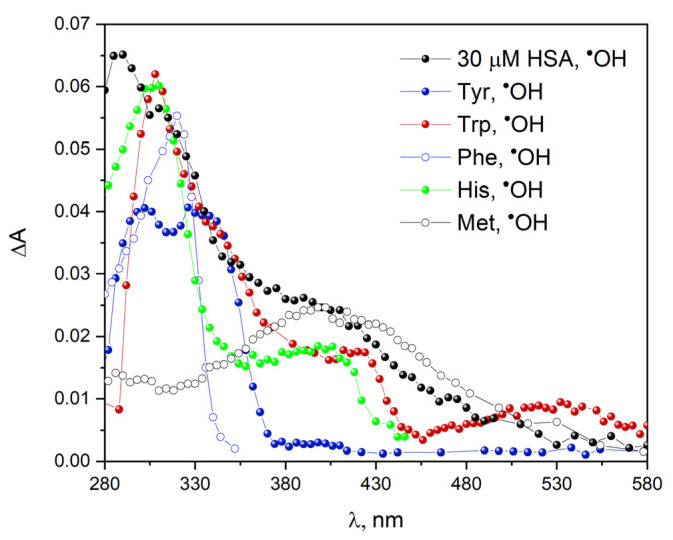
Transient absorption spectra of N_2_O-saturated buffer solutions containing 30 μM HSA, 1 mM Tyr, 1 mM Trp, 1 mM His, 1 mM Phe, and 1 mM Met, obtained for an irradiation dose of 800 Gy. Each of the above-mentioned solutions was subjected to radiolysis separately.

## Data Availability

Not applicable.

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
