# Peer review of "Spontaneous and Ionizing Radiation-Induced Aggregation of Human Serum Albumin: Dityrosine as a Fluorescent Probe"

_ijms, 2022, doi:10.3390/ijms23158090_

Round 1

Reviewer 1 Report

In the manuscript Authors report on the radiolysis of albumin under oxidative condition. The main aim of the study is to investigate the reliability of the application of fluorescence detection of intermolecular dityrosine bridges (obtained as a result of the attack of oxidizing radicals on HSA) to assess the degree of radiation damage to biomolecules. The subject of the article is interesting an in line with scientific profile of Special Issue of International Journal of Molecular Sciences.
The publication explains in detail the interesting phenomenon of spontaneous aggregation of human blood plasma albumin. The consequence of aggregation is the formation of HSA nanostructures, which are characterized by the emission of blue light in the spectral range unusual for this protein. It has been shown that irradiation of HSA solutions under oxidative stress conditions results in the formation of stable protein aggregates. The authors showed that depending on the nature of the radicals reacting with proteins, two different types of HSA aggregates are formed. Pulse radiolysis of HSA solutions confirmed that the reaction of mild oxidants (Br2•⁃, N3•, SO4•⁃) with albumin leads to the formation of covalent bonds between tyrosine residues. In the case of •OH radicals, different covalent aggregates of HSA are formed. In my opinion, the greatest advantage of the manuscript is the explanation that the application of the dityrosine fluorescence detection protocol, which has been known for many years, is correct, but requires modification in order to quantify damage to albumin.
The drawings at work are generally well described, but the authors should supplement the captions with details, e.g. excitation wavelengths. The lack of a detailed description requires confronting the text of the manuscript with the caption for the drawing. For example, Figure 16 shows two vertical lines, and it is unclear what they are for. The excitation wavelength information is given in the figure but should also be included in the caption. The manuscript is well informed in the relevant field but in my opinion one paper should be included in References. I miss a reference to a recently published work [1], which was carried out at the Institute where the authors of the manuscript are employed. The aforementioned publication [1] concerns the mechanical approach to the formation of dityrosine in proteins by ionizing radiation, and the authors of a peer-reviewed manuscript should refer to it briefly.

          To conclude, I recommend a publication of the manuscript: Spontaneous and ionizing radiation-induced aggregation of human serum albumin. Dityrosine as a fluorescent probe by Radomska and Wolszczak in Special Issue "Radiation and Photochemical Modifications in Proteins: Mechanistic Aspects and Applications" after some corrections according to comments given above.

 1. A mechanistic approach towards the formation of bityrosine in proteins by ionizing radiation – GYG model peptide
Sowinski S., Varca G.H.C, Kadlubowski S., Lugao A. B. and Ulanski P.
Rad. Phys. Chem. 188 (2021) 109644.

Author Response

As suggested by the reviewer, we checked all Captions and improved some of them. All text changes are marked up using the Track changes function. We would like to thank the reviewer for the suggestion to include in the manuscript a reference [76] on the mechanical approach to the formation of dityrosine in proteins by ionizing radiation. We are currently preparing a publication on the formation of protein aggregates (ovalbumin, lactoglobulin, gamma globulin, HSA, BSA and papain) as a result of irradiation. In this next article, we plan to carefully respond to the content of the work [76]. It's easy because we cooperate with the authors of the publication. In the present manuscript we mention [76] in the block of works related to the technological platform based on radiation-cross-linked proteins and enzymes for the production of materials of biomedical importance (References ).
[76] Sowinski S., Varca G.H.C.; Kadlubowski S.; Lugao A. B.; Ulanski P. A mechanistic approach towards the formation of bityrosine in proteins by ionizing radiation–GYG model peptide. Rad. Phys. Chem. 2021, 188, 109644.

Reviewer 2 Report

The authors state the formation of dimers and aggregates. Is it possible to outline the difference? 

Also, the author pointed out some dimers that are resultant form covalent bonds. What happens under irradiation? Where the aggregates found irreversible? 

Recent work has been performed on the dityrosine formation in papain, and some of the mechanisms have also been detailed. A technological paltform has been developed based on radiation crosslinked proteins. Perhaps such information could be a valuable addition to the file, despite the differences between such molecules, in the sense of outlining the technological applications of such aggregates.

Also, other author have explored the formation of BSA aggregates in absence and presence of ethanol and scavengers. The addition of some of those works, could help the audience understand the the relevance of the present paper in line with recent technological findings. 

Author Response

Our (quite old) HPLC measurements and electrophoresis studies of HSA solutions as well as analogous experiments described in the literature indicate the spontaneous formation of dimers. These dimers are formed even at low concentrations of HSA (10 μM) and are dynamic species, they reversibly dissociate with dilution of the solution. As the protein concentration increases, trimers, tetramers, pentamers, etc. appear. We called these higher-than-dimeric forms of HSA complexation spontaneous aggregates (We have included this sentence on page 6 All text changes are marked up using the Track changes function). Depending on the nature of the protein-reactive radicals, two different types of HSA aggregates are formed. Aggregates produced by irradiation are irreversible. These two types of aggregates (for small doses of ionizing radiation mainly dimers are formed) lead to the formation of covalent bonds (We have included this two sentence in the page 32). Pulse radiolysis of HSA solutions confirmed that the reaction of mild oxidants (Br2•−, N3•, SO4•−) with albumin leads to the formation of covalent bonds between tyrosine residues. The formation of dityrosine bridges is easily monitored by fluorescence detection after excitation at 315-325 nm. In the case of reactive oxidants (•OH radicals and partly Cl2•−), different covalent aggregates of HSA are formed. This leads to the formation of numerous covalent bridges (including C-C, it is also possible to generate C-S bonds and dityrosine isomers which are non-fluorescent), and consequently HSA dimers and aggregates. It was evident that HSA aggregates increase in size after irradiation, regardless of the type of oxidant. The SDS-PAGE date indicates that irradiation of HSA solution leads to generation of aggregates with high MW (this aggregates are not able to enter in the running gel). We are currently preparing a publication on the formation of protein aggregates (ovalbumin, lactoglobulin, gamma globulin, HSA, BSA and papain) for publication. In this study, we will also describe the formation of aggregates under reducing conditions. We are grateful to the reviewer for drawing our attention to the technological platform for obtaining biomedically important materials based on radiation cross-link proteins and enzymes. Papain is the main subject of our research which will be described soon. We want to make a short comment in this manuscript (page 33-34):
Recently, a technological platform based on radiation cross-linked proteins and enzymes has been developed to obtain materials of biomedical importance. In our upcoming publications, we intend to address important mechanistic aspects of synthesis by irradiation proteins and enzymatic aggregates in line with the latest technological findings [6, 60,73-76].
[6] Queiroz, R.G.; Varca, G.H.C.; Kadlubowski, S.; Ulanski, P.; Lugão, A.B. Radiation synthesized protein-based drug carriers: size-controlled BSA nanoparticles. Int. J. Biol. Macromol. 2016, 85, 82–91.
[73] Fazolin, G.N.; Varca, G.H.C.; de Freitas, L.F.; Rokita, B.; Kadlubowski, S.; Lugão, A.B. Simultaneous intramolecular crosslinking and sterilization of papain nanoparticles by gamma radiation. Radiat. Phys. Chem. 2020, 171, 108697.
[74] Fazolin, G.N.; Varca, G.H.C.; Kadlubowski, S.; Sowinski, S.; Lugão, A.B. The effects of radiation and experimental conditions over papain nanoparticle formation: Towards a new generation synthesis. Radiat. Phys. Chem. 2020,169, 107984
[75] Varca, G.H.C.; Kadlubowski, S.; Wolszczak, M.; Lugão, A.B.; Rosiak, J.M.; Ulanski, P. Synthesis of papain nanoparticles by electron beam irradiation–a pathway for controlled enzyme crosslinking. Int. J. Biol. Macromol. 2018, 92, 654–659. [76] Sowinski S., Varca G.H.C, Kadlubowski S., Lugao A. B. and Ulanski P. A mechanistic approach towards the formation of bityrosine in proteins by ionizing radiation–GYG model peptide. Rad. Phys. Chem. 2021, 188, 109644.